# Horizontal acquisition of prokaryotic hopanoid biosynthesis reorganizes membrane physiology driving lifestyle innovation in a eukaryote

Bhagyashree Dasari Rao[1,2,4], Elisa Gomez-Gil [1,2,4], Maria Peter[3], Gabor Balogh [3], Vanessa Nunes [2], James I. MacRae [2], Qu Chen [2], Peter B. Rosenthal [2] & Snezhana Oliferenko [1,2] ✉

Horizontal gene transfer is a source of metabolic innovation and adaptation to new environments. How new metabolic functionalities are integrated into host cell biology is largely unknown. Here, we probe this fundamental question using the fission yeast *Schizosaccharomyces japonicus*, which has acquired a squalene-hopene cyclase Shc1 through horizontal gene transfer. We show that Shc1-dependent production of hopanoids, mimics of eukaryotic sterols, allows *S. japonicus* to thrive in anoxia, where sterol biosynthesis is not possible. We demonstrate that glycerophospholipid fatty acyl asymmetry, prevalent in *S. japonicus*, is crucial for accommodating both sterols and hopanoids in membranes and explain how Shc1 functions alongside the sterol biosynthetic pathway to support membrane properties. Reengineering experiments in the sister species *S. pombe* show that hopanoids entail new traits in a naïve organism, but the acquisition of a new enzyme may trigger profound reorganization of the host metabolism and physiology.

The physicochemical properties of biological membranes are fine-tuned to support cellular and organismal physiology. Deregulation of lipid composition may change membrane physicochemical features, affecting the functions of membrane-associated proteins, cell physiology, and survival[1,2]. Eukaryotic membranes include three major lipid classes: glycerophospholipids (GPLs), sphingolipids and sterols.

Sterols are essential components of canonical eukaryotic membranes, maintaining the structure, fluidity, and permeability of the lipid bilayer[3]. Although the sterol biosynthetic pathway is evolutionarily conserved in eukaryotes, it yields different final products in animals (cholesterol), plants (phytosterols), and fungi (ergosterol).

Crucially, de novo sterol biosynthesis requires multiple oxygen-dependent steps and thus, eukaryotic membrane chemistry relies heavily on oxygen availability[4]. However, many eukaryotes inhabit or explore hypoxic and anoxic niches[5], which presumably entails the evolutionary adaptation of their lipid metabolism to maintain membrane integrity in oxygen-deprived environments. Unlike the popular model fission yeast *Schizosaccharomyces pombe*, which is an obligate aerobe, its relative *Schizosaccharomyces japonicus* thrives in both aerobic and anaerobic environments[6-8]. The *S. japonicus* lineage has diverged from *S. pombe* ~200 million years ago[9], and has lost the ability to respire oxygen, instead becoming a committed fermenting species[10]. *S. japonicus* also has an expanded optimum temperature range, growing at temperatures up to 42 °C, unlike *S. pombe* that is restricted to temperatures below 36 °C[8]. *S. japonicus* is a dimorphic organism, capable of burrowing deep into the substrate in its hyphal form[11,12], presumably reaching beyond the limit of oxygen diffusion. Unlike budding yeast that grows well anaerobically only when

[1]Randall Centre for Cell and Molecular Biophysics, School of Basic and Medical Biosciences, King's College London, Guy's Campus, London, UK. [2]The Francis Crick Institute, London, UK. [3]Institute of Biochemistry, HUN-REN Biological Research Centre, Szeged, Hungary. [4]These authors contributed equally: Bhagyashree Dasari Rao, Elisa Gomez-Gil. ✉e-mail: snezhana.oliferenko@kcl.ac.uk

provided with ergosterol[13], S. *japonicus* does not require supplementation for anaerobic growth, suggesting that it has evolved a strategy to circumvent the oxygen demands of lipid biosynthesis.

Many prokaryotes use hopanoids as sterol surrogates to regulate membrane properties under harsh conditions[14,15]. Sterols and hopanoids are derived from the same squalene precursor, but hopanoids are cyclized from squalene in a single, oxygen-independent step catalyzed by a squalene-hopene cyclase (SHC)[15]. Interestingly, phylogenetic analyses show multiple independent instances of horizontal gene transfer of SHC and SHC-related genes into eukaryotic lineages[6,16,17]. S. *japonicus* is known to synthesize hopanoids even in normoxia and upregulate hopanoid synthesis upon oxygen depletion[6]. It was suggested that hopanoids were synthesized by an SHC originating from an *Acetobacter*-related species[6], although the role of this enzyme in S. *japonicus* physiology has not been tested.

Here, we explore the contribution of the SHC to S. *japonicus* lifestyle and explain how it functions alongside the ergosterol biosynthetic pathway to support membrane properties both in the presence and in the absence of oxygen. Using biophysical approaches and cryo-EM, we identify the glycerophospholipid fatty acyl asymmetry as a key feature allowing both native and foreign triterpenoids to function in S. *japonicus* membranes. Finally, by engineering the related fission yeast S. *pombe*, we explore how hopanoids can be integrated into the logic of ergosterol-rich eukaryotic physiology.

## Results

### Both ergosterol and hopanoids support S. *japonicus* lifestyle

S. *japonicus* genome harbors a horizontally transferred gene encoding a SHC protein (SJAG_03360, hereafter referred to as Shc1)[6,18]. To synthesize hopanoids, Shc1 is predicted to use squalene, which is also a precursor for the squalene epoxidase Erg1, a rate-limiting enzyme of ergosterol biosynthesis. Unlike Erg1 and several other enzymes in the ergosterol pathway, which require oxygen for their function, SHCs synthesize hopanoids in an oxygen-independent manner[19] (Fig. 1a). Shc1 tagged at its endogenous locus with superfolder GFP (Shc1-sfGFP) localized to cellular membranes, including the nuclear envelope (NE), cortical endoplasmic reticulum (ER) and/or the plasma membrane, and the vacuolar membranes (Fig. 1b). Unlike the wild-type or Shc1-sfGFP-expressing cells, S. *japonicus* mutant lacking *shc1* was unable to grow in the absence of oxygen (Fig. 1c), demonstrating the requirement for Shc1 in S. *japonicus* anaerobic growth.

Sterols are thought to be crucial for the maintenance of biophysical properties of eukaryotic lipid bilayers[20]. To probe whether hopanoids can act as ergosterol substitutes in S. *japonicus* membranes in normoxia, we treated cells with the Erg1 inhibitor terbinafine, which disrupts sterol biosynthesis[21] (Fig. 1a). Contrary to the S. *japonicus* wild type, the *shc1Δ* mutant exhibited a marked growth sensitivity to terbinafine. As expected, a related species S. *pombe*, which does not harbor an SHC[9], was as sensitive to the drug as the S. *japonicus shc1Δ* strain (Fig. 1d).

Since sterols are critical components of canonical eukaryotic membranes, the *erg1* gene is essential for life in S. *pombe* and other fungi[22–24]. Strikingly, we were able to generate S. *japonicus erg1Δ* mutant, suggesting that this fission yeast does not strictly require sterols to support its growth. As expected, *erg1Δ* cells were not stained by the naturally fluorescent sterol marker polyene filipin[25] (Supplementary Fig. 1a), and exhibited resistance to amphotericin B, a drug that induces the formation of pores in the plasma membrane by binding ergosterol[26] (Supplementary Fig. 1b). We were not able to recover double *erg1Δ shc1Δ* mutants in the genetic crosses between single *erg1Δ* and *shc1Δ* mutant strains (0/47 expected progeny; see "Methods"). Together, our genetic data, drug treatment, and anaerobic growth assays confirm that S. *japonicus* relies on Shc1 activity in the absence of sterol biosynthesis.

To understand how the lack of Shc1 or Erg1 impacted membrane lipid composition, we performed gas chromatography-mass spectrometry (GC-MS) and shotgun electrospray ionization mass spectrometry (ESI-MS) analyses of S. *japonicus* total lipid extracts. As previously described[6], S. *japonicus* wild-type cells produced high levels of hopanoids even in the presence of oxygen, with diplopterol being the most abundant. We also detected the hopenes hop-22(29)-ene and hop-17(21)-ene (Fig. 1e and Supplementary Data 1a).

The S. *japonicus shc1Δ* mutant did not produce hopanoids, demonstrating that Shc1 is indeed responsible for hopanoid synthesis in this organism. As expected, we did not detect ergosterol in *erg1Δ* cells grown in normoxia or in the wild-type cells grown in anaerobic environment. Interestingly, whereas the cellular amount of hopanoids increased profoundly when ergosterol synthesis was inhibited either by *erg1Δ* mutation or the growth in anoxia, the converse was not true. Cells lacking Shc1 and wild-type cells exhibited comparable ergosterol content (Fig. 1e and Supplementary Data 1a). The amount of squalene, the substrate for both Shc1 and Erg1, dropped dramatically in the absence of oxygen (Fig. 1e).

Notably, the lack of either hopanoids or ergosterol led to changes in the total cellular lipid landscape, as shown by the SoamD score, which represents the sum of absolute mol% difference relative to the wild type for every lipid species[27] (Supplementary Fig. 1c). In cells lacking Shc1, the ratio between phosphatidylethanolamine (PE) and the sum of phosphatidylcholine and phosphatidylinositol (PE/(PC + PI)) was significantly reduced, suggesting a cellular response to stabilize the membranes[28] (Fig. 1f and Supplementary Fig. 1d). We also detected decreased abundance of lysophosphatidylethanolamine (LPE) and ceramide (Cer), and increased levels of phosphatidic acid (PA), diacylglycerol (DG) and inositol phosphoceramide (IPC) (Supplementary Fig. 1e).

The cellular PE/(PC + PI) ratio was similarly decreased when cells were unable to synthesize sterols, either in the *erg1Δ* mutant or when grown in the absence of oxygen (Fig. 1f and Supplementary Fig. 1d). In addition, the level of FA chain desaturation, already low in S. *japonicus*[29], decreased further in *erg1Δ* mutant (Fig. 1g and Supplementary Fig. 1g, h), and the average FA chain length mildly increased (Supplementary Fig. 1f). The decrease in PE/(PC + PI) ratio, decrease in FA desaturation and an increase in GPL chain length might all contribute to membrane stability in the absence of sterols[28]. As expected, FA desaturation was virtually absent in anaerobic conditions, due to the oxygen dependence of the delta-9 desaturase Ole1[30], with cells producing higher amounts of asymmetrical GPL species (Fig. 1g and Supplementary Fig. 1g, h). We also detected differences in the abundance of some of the minor GPL and lysoglycerophospholipid classes, sphingolipids, and storage lipids in both *erg1Δ* mutant and the wild type growing in anoxia (Supplementary Fig. 1e and Supplementary Data 1a).

To assess how hopanoids and ergosterol contribute to S. *japonicus* physiology, we investigated cellular growth and survival of mutants lacking either Shc1 or Erg1, both in the rich Yeast Extract with Supplements (YES) and in the modified minimal Yeast Nitrogen Base (YNB) media (see "Methods"). Fission yeast cells grow at cell tips and divide in the middle, exhibiting stereotypic pill-shaped geometry in exponentially growing cultures[31]. In minimal medium, where the anabolic demands are high, S. *japonicus* grows slower and undergoes pronounced downscaling of its geometry, dividing at decreased length and width but maintaining its aspect ratio[32]. S. *japonicus* cells lacking *shc1* exhibited somewhat perturbed geometry, dividing at increased cell length. Cell width was also deregulated (Fig. 1h–j). Despite differences in cell geometry, *shc1Δ* cultures grew at normal rates at 30 °C (Fig. 1k, l). Interestingly, *shc1Δ* cells showed decreased viability at 40 °C in the minimal medium (Supplementary Fig. 1i). Additionally, cells lacking Shc1 exhibited mild defects in survival and regrowth from the stationary phase (Fig. 1m, n and Supplementary Fig. 1j). The lack of Erg1

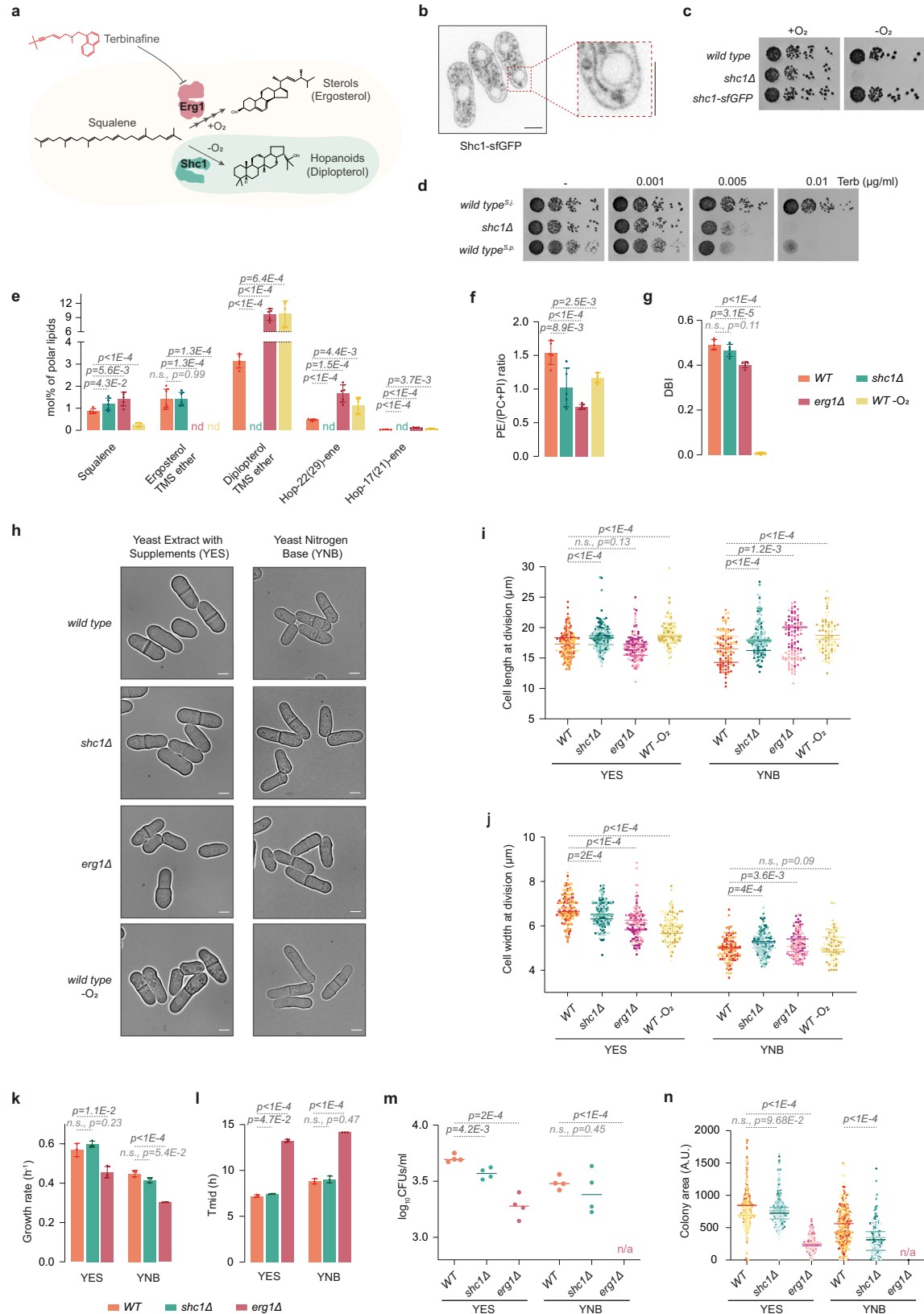

led to more pronounced phenotypes. At the optimal temperature of 30 °C, *erg1Δ* cells exhibited defects in cell geometry (Fig. 1h–j). Interestingly, wild type cells grown in the absence of oxygen, where ergosterol biosynthesis is inhibited, also showed deregulation of cellular geometry, as compared to aerobic growth (Fig. 1h–j). Cells lacking

Erg1 exhibited slower growth rate and extended lag-phase (Fig. 1k, l), and were less viable at higher temperatures in the minimal medium (Supplementary Fig. 1i). Cell survival in stationary phase and subsequent regrowth were strongly affected by the absence of sterols, with the *erg1Δ S. japonicus* mutant not being able to form colonies in

**Fig. 1 | *S. japonicus* relies on hopanoid synthesis to survive in the absence of ergosterol. a** Squalene utilization in *S. japonicus* via oxygen-dependent ergosterol biosynthesis pathway starting with squalene epoxidase Erg1 or oxygen-independent hopanoid pathway via squalene-hopene cyclase Shc1. Terbinafine inhibits Erg1. **b** Spinning disk confocal image of Shc1-sfGFP-expressing cells including a magnified view. **c** Serial dilution assay of *S. japonicus* strains of indicated genotypes in normoxia or anoxia in YES medium. **d** Serial dilution assay of *S. japonicus WT*, *shc1Δ*, and *S. pombe WT* strains in normoxia in YES medium with or without terbinafine (Terb). **e** GC-MS quantification of triterpenoid content in *S. japonicus WT*, *shc1Δ*, and *erg1Δ* cells in normoxia, and *WT* in anoxia. nd, not detected. **f** Ratio of phosphatidylethanolamine (PE) to phosphatidylcholine (PC) + phosphatidylinositol (PI). **g** Double-bond indexes (DBI) for PC, PI, PE, and phosphatidylserine (PS) in *S. japonicus* strains of indicated genotypes and conditions. **h** Micrographs of *S. japonicus WT*, *shc1Δ*, and *erg1Δ* cells in normoxia, and *WT* in anoxia. Quantification of (**i**) cell length and (**j**) width at division in *S. japonicus WT* (*n* = 151 cells in YES; *n* = 110 cells in YNB), *shc1Δ* (*n* = 155 cells in YES; *n* = 142 cells in YNB), and *erg1Δ* (*n* = 129 cells in YES; *n* = 104 cells in YNB) in normoxia, and *WT* (*n* = 121 cells in YES; *n* = 92 cells in YNB) in anoxia. Bars represent medians (*n* = 3 biological repeats). **k** Growth rates of *WT*, *shc1Δ*, and *erg1Δ* in indicated media. **l** Time to reach half of the maximum population (Tmid) in cultures from (**k**). **m** Survival of *S. japonicus* cells of indicated genotypes in stationary phase. **n** Colony area of CFUs from (**m**) for *S. japonicus WT* (*n* = 506 colonies in YES; *n* = 307 colonies in YNB), *shc1Δ* (*n* = 373 colonies in YES; *n* = 274 colonies in YNB), and *erg1Δ* (*n* = 193 colonies in YES; *n* = 0 colonies in YNB). **b, h** Scale bars: 5 μm. **e–g** Data shown as average ± S.D. (*n* = 3 biological, 2 technical repeats). **k, l** Data shown as average ± S.D. (*n* = 3 biological repeats). (**m, n**) Bars show medians (*n* = 4 biological repeats). (**e–g, i–n**) p-values from two-tailed unpaired *t*-test. Source data provided as a Source Data file.

the minimal medium (Fig. 1m, n and Supplementary Fig. 1j). Thus, it appears the aerobic physiology of *S. japonicus* relies in larger part on Erg1 rather than Shc1.

Taken together, our results suggest that ergosterol and hopanoids collaborate to support membrane properties in *S. japonicus* in a variety of physiological situations. Importantly, the overall lipidome of this organism has been adapted to the acquisition of hopanoid biosynthesis through horizontal gene transfer.

## Asymmetrical saturated lipids can use either hopanoids or ergosterol to support membrane properties

To understand how *S. japonicus* may rely on either ergosterol or hopanoids to support its physiology, we turned to a bottom-up approach relying on model membranes. We used PC(C18:0/C10:0) (1-stearoyl-2-decanoyl-sn-glycero-3-phosphocholine, SDPC) and either PC(C16:0/C18:1) (1-palmitoyl-2-oleoyl-sn-glycero-3-phosphocholine, POPC) or PC(C18:1/C18:1) (1,2-dioleoyl-sn-glycero-3-phosphocholine, DOPC) as models for asymmetrical saturated *S. japonicus*- or symmetrical unsaturated *S. pombe*-like glycerophospholipids, respectively (Fig. 2a and ref. 29).

When combined with 30 mol% of either ergosterol or diplopterol, all three glycerophospholipids formed single-phase membranes, as indicated by the labeling of giant unilamellar vesicles (GUVs) with FASTDiI[33,34], a dye that in the context of phase-separated membranes partitions preferentially into the liquid-disordered (Ld) membrane phase (Supplementary Fig. 2a, b). Suggesting lower membrane order, membranes formed with the asymmetrical glycerophospholipid SDPC in single or two-component mixtures with either triterpenoid exhibited higher water permeability than those made with the symmetrical unsaturated POPC (Supplementary Fig. 2c, d).

We estimated membrane order in two-component single-phase membranes by measuring the anisotropy of the rod-shaped fluorophore 1,6-diphenyl-1,3,5-hexatriene (DPH). DPH fluorescence anisotropy depends on its rotational mobility in the bilayer. It is more restricted in ordered environments, producing higher anisotropy values[35,36]. As expected, the control gel-like large unilamellar vesicles (LUVs) made of DPPC and ergosterol showed the highest anisotropy. Ergosterol promoted higher order in the context of all two-component lipid mixtures, as compared to diplopterol (Supplementary Fig. 2e). To corroborate our spectroscopy data, we imaged GUVs made from these two-component lipid mixtures using the fluorescent probe C-laurdan (structure in inset of Fig. 2b). The emission properties of this probe depend on the lipid bilayer environment and can be used to estimate relative levels of lipid packing by calculating a parameter called Generalized Polarization (GP)[37–39]. The membranes composed of the gel-forming DPPC and ergosterol displayed the maximum blue-shifted peak (~453 nm) and the highest GP, indicating that they were indeed highly ordered (Fig. 2b, and Supplementary Fig. 2f, g). In two-component mixtures containing either symmetrical unsaturated POPC or asymmetrical saturated SDPC, ergosterol supported higher

membrane order as compared to diplopterol (Fig. 2b), consistent with our DPH anisotropy measurements.

Mixtures of the unsaturated GPLs with the gel-forming symmetrical saturated glycerophospholipid PC(C16:0/C16:0) (1,2-dipalmitoyl-sn-glycero-3-phosphocholine, DPPC) and sterols tend to separate into the liquid-disordered (Ld) and liquid-ordered (Lo) phases in vitro. Lipid-lipid interactions resulting in macroscopic phase separation in model membranes and plasma membrane-derived vesicles are thought to contribute to the generation and/or stabilization of lateral heterogeneities in vivo, critical for the function of biological membranes[2]. Importantly, both *S. pombe*-like symmetrical unsaturated glycerophospholipids, POPC or DOPC, exhibited phase separation in the presence of 20 mol% ergosterol but not diplopterol (Fig. 2c).

We then explored the phase behavior in three-component liposomes containing the *S. japonicus*-like asymmetrical SDPC glycerophospholipid. SDPC did not support phase separation when combined with the unsaturated symmetrical POPC, further indicating that it conferred disorder to membranes. Strikingly, either 20 mol% ergosterol or 20 mol% diplopterol was sufficient to induce phase separation in liposomes containing SDPC together with the gel-forming DPPC (Fig. 2d). Of note, ergosterol was more efficient in promoting phase separation in model membranes containing the asymmetrical GPL (Fig. 2d).

Our results so far indicated that asymmetrical saturated lipids are disordered but can accommodate either ergosterol or diplopterol to promote phase separation in model membranes. To measure membrane packing in both phases of all phase-separated three-component membranes, we estimated the GP values using C-laurdan. Of note, whereas diplopterol indeed supported separation of lipids into Lo and Ld phases in GUVs containing SDPC and DPPC, the GP values were lower for the Lo phase as compared to GUVs containing ergosterol (Fig. 2e–g and Supplementary Fig. 2h).

To understand the roles of both ergosterol and hopanoids in sustaining membrane order in vivo, we carried out GP measurements of live fission yeast cells labeled with C-laurdan[40,41]. Both plasma membrane and nuclear membrane order in *S. japonicus* cells was higher as compared to the sister species *S. pombe*, in agreement with our earlier measurements using di-4-ANEPPDHQ dye[29]. Importantly, the absence of either hopanoids (*shc1Δ*) or ergosterol (*erg1Δ*) led to a comparable decrease in membrane order, both at the plasma membrane and the nuclear membrane (Fig. 2h, i).

Overall, our model membrane data suggest that ergosterol is more efficient than diplopterol at promoting membrane order. However, both triterpenoids can facilitate phase separation in membranes containing the asymmetrical saturated glycerophospholipids abundant in *S. japonicus*. In contrast, only ergosterol can achieve this crucial effect in membranes composed of symmetrical *S. pombe*-like glycerophospholipids. Importantly, in live *S. japonicus* cells, both hopanoids and ergosterol contribute to maintaining membrane order to a comparable extent.

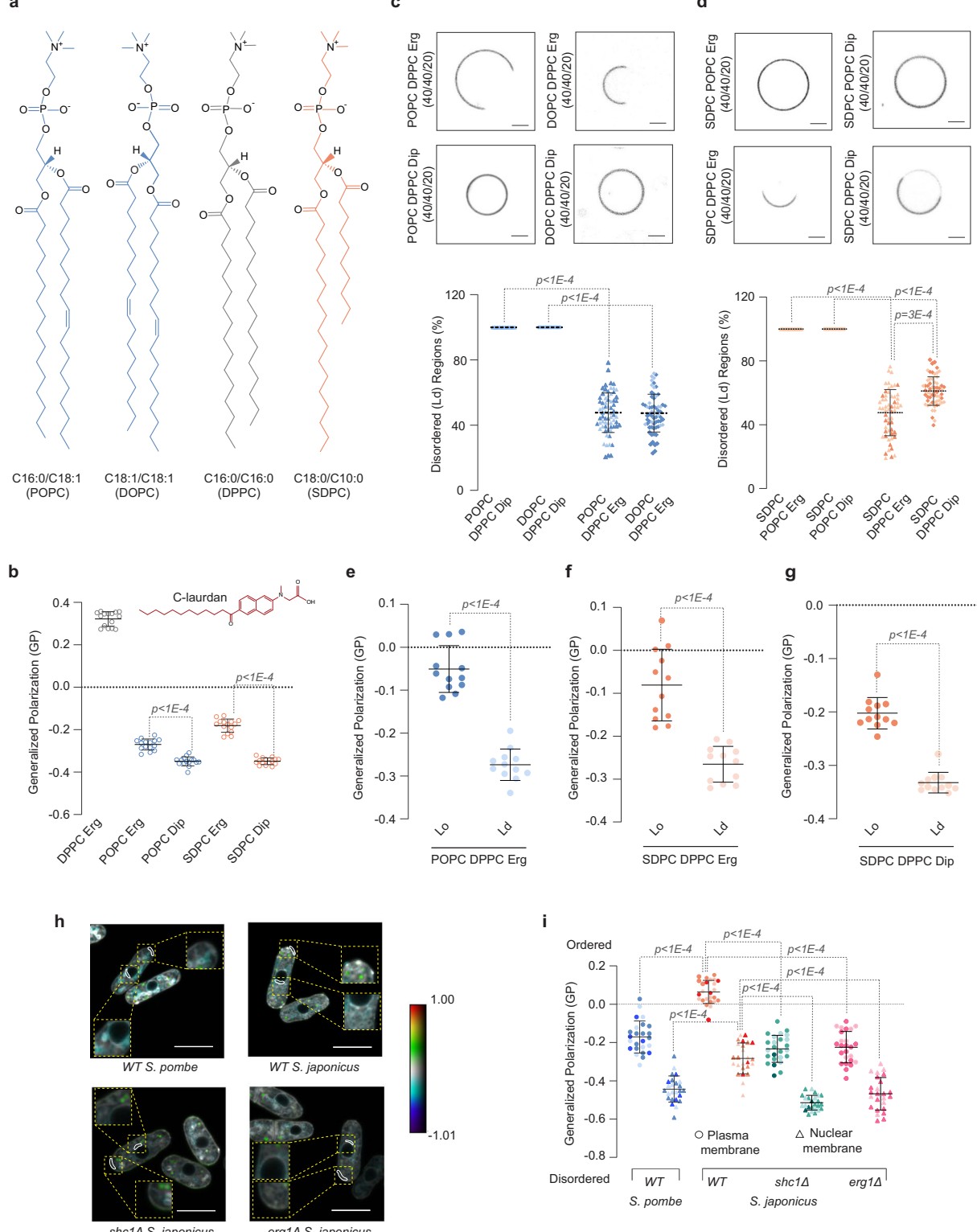

**Cryo-EM imaging demonstrates that asymmetrical glycerophospholipids form thinner membranes and support membrane phase separation in the presence of either ergosterol or diplopterol**

To directly image our model membranes, estimating membrane thickness and exploring their phase behavior, we employed cryogenic electron microscopy (cryo-EM), which has been successfully used to detect nanoscopic domains in synthetic and bioderived membranes[42,43]. Membrane thickness can be estimated by $D_{TT}$, a

distance between two troughs in intensity across the bilayer, which correspond to electron-rich head group regions in glycerophospholipids. Strikingly, single-component membranes made of saturated asymmetrical SDPC were considerably thinner ($D_{TT}$ = 27.78 nm ± 1.72 nm, $n$ = 20) as compared to those composed of symmetrical unsaturated POPC ($D_{TT}$ = 36.39 nm ± 2.83 nm, $n$ = 20) (Fig. 3a, b).

We observed a similar trend in two-component lipid mixtures. Whereas the control gel-like liposomes (DPPC with ergosterol)

**Fig. 2 | Asymmetrical glycerophospholipids support membrane phase properties both in the presence of ergosterol and diplopterol. a** Chemical structures of glycerophospholipids used in the study. **b** Estimation of membrane order from spectral GP imaging of two-component GUVs made with POPC or SDPC with either ergosterol or diplopterol. Control values for gel-like DPPC/Erg GUVs are shown for reference. The values represent average ± S.D. ($n = 15$ GUVs, one biological repeat), estimated using Eq. (2). The top panel shows representative spinning disk confocal images (mid-plane) of three-component GUVs assembled with equimolar amounts of symmetrical unsaturated glycerophospholipids POPC or DOPC and the gel-like DPPC (**c**), or asymmetrical saturated glycerophospholipid SDPC (**d**) with either 20 mol% ergosterol or diplopterol. GUVs were labeled with FAST DiI. Scale bars represent 2 μm. Dot plots depict the quantification of area of liquid-disordered (Ld) regions upon phase separation (average ± S.D., with the following sample sizes: POPC DPPC Dip ($n = 20$ GUVs), DOPC DPPC Dip ($n = 20$ GUVs), POPC DPPC Erg ($n = 71$ GUVs), DOPC DPPC Erg ($n = 66$ GUVs) SDPC POPC Erg ($n = 20$), SDPC POPC Dip ($n = 20$), SDPC DPPC Erg ($n = 58$) and SDPC DPPC Dip ($n = 60$), from two biological repeats). **e–g** Quantification of order in two phases of three-component GUVs made with the following lipids (40/40/20 mol%): POPC, DPPC, and ergosterol (**e**), SDPC, DPPC and ergosterol (**f**), and SDPC, DPPC and diplopterol (**g**). Lo represents the liquid-ordered regions, and Ld represents liquid-disordered regions. GP values were calculated using Eq. (2). Average ± S.D. are shown ($n = 12$ GUVs, one biological repeat). **h** Representative single-plane pseudo-colored generalized polarization (GP) images. Color bar indicates the range of GP values where blues show low membrane order and reds show high membrane order. Areas quantified are outlined in white. Zoomed regions show plasma membrane and nuclear membrane areas used to quantify GP values. Scale bar, 5 μm. **i** A plot representing individual GP values quantified at the plasma membrane regions at cell tips and at the nuclear membrane in cells of indicated genotypes. Average ± S.D. are shown ($n = 25$ cells from three biological repeats). (**b–g, i**) $p$ values were obtained by two-tailed unpaired parametric $t$-tests. Source data are provided as a Source Data file.

exhibited the highest $D_{TT}$, both ergosterol- and diplopterol-containing SDPC bilayers were thinner than their counterparts made of symmetrical unsaturated POPC. Consistent with its greater ordering potential, ergosterol increased the thickness of SDPC-containing membranes (Fig. 3c, see quantification in Fig. 3f and Supplementary Fig. 3).

Using cryo-EM, we were not able to detect coexisting membrane domains in three-component phase-separated *S. pombe*-like liposomes (POPC/DPPC/ergosterol), likely due to relatively small differences in membrane thickness between DPPC and POPC (Fig. 3d, quantification in Fig. 3f and Supplementary Fig. 3). Remarkably, we observed a clear phase separation within single liposomes in three-component mixtures made with asymmetrical *S. japonicus*-like SDPC glycerophospholipid (Fig. 3d–f). The width measurements of thick (ordered) and thin (disordered) domains were comparable in SDPC membranes containing either ergosterol or diplopterol (Fig. 3d–f).

Our cryo-EM data indicate that asymmetrical saturated glycerophospholipids form thinner membranes, in line with shortening of transmembrane helices in the *S. japonicus* orthologues of several proteins found at early Golgi and ER-organelle contact sites[29]. The remarkably similar hydrophobic thickness values for the ordered and disordered domains in asymmetrical saturated glycerophospholipid-based phase-separated model membranes containing either ergosterol or diplopterol suggest that in vivo, the asymmetrical GPLs could accommodate similar sets of proteins and functions, regardless of the triterpenoid type.

## Hopanoid synthesis in *S. pombe* enables new physiological features

Our results so far suggested that eukaryotic lipidomes must be adapted to accommodate the bacterial sterol mimics hopanoids. *S. japonicus* appears to solve this problem at least in part by maintaining high levels of FA asymmetry in glycerophospholipids, which is a minor feature in the lipidomes of its relative *S. pombe* and budding yeast[29,44]. We wondered if introducing Shc1 into a naïve organism, such as the sister species *S. pombe*, could lead to changes in cellular lipid landscape and provide immediate benefits for adapting to new environments.

To this end, we integrated a construct encoding the *S. japonicus* Shc1 tagged with sfGFP as a single copy under the control of a strong, constitutive *tdh1* promoter (*ptdh1:shc1^{S.j.}-sfGFP*) in the *S. pombe* genome. In *S. pombe*, Shc1^{S.j.}-sfGFP localized predominantly to the NE and cortical ER and/or the plasma membrane, reminiscent of its localization in *S. japonicus* (Fig. 4a). Triterpenoid analysis by GC-MS showed that *S. pombe* cells expressing Shc1^{S.j.} synthesized diplopterol (Fig. 4b), although its total cellular amount was considerably lower than in *S. japonicus* ($0.266 \pm 0.035$ mol% polar lipids in *S. pombe* vs $3.134 \pm 0.311$ in *S. japonicus*). We did not detect two minor hopanoid species present

in *S. japonicus* – hop-22(29)-ene and hop-17(21)-ene–in Shc1^{S.j.}-expressing *S. pombe* (Supplementary Data 1b)

Enabling hopanoid synthesis in *S. pombe* significantly altered the overall cellular lipidome. Although ergosterol levels were not greatly affected (Fig. 4b), we observed differences in the abundance of major GPL classes (Supplementary Fig. 4a), resulting in a significant decrease in the cellular PE/(PC + PI) ratio (Fig. 4c). We also detected major changes in the abundance of lysoglycerophospholipids, sphingolipids and storage lipids in *ptdh1:shc1^{S.j.}-sfGFP S. pombe* cells as compared to the wild type (Supplementary Fig. 4b).

Membrane glycerophospholipids in *S. pombe* largely consist of symmetrical di- and mono-unsaturated species [36:2] and [34:1], although *S. pombe* also synthesizes some asymmetrical saturated [26:0] and [28:0] GPLs[29,45]. Interestingly, we observed an increase in the [36:2] GPLs with a concomitant decrease in [26:0] and [28:0] species in Shc1^{S.j.}-expressing cells (Supplementary Fig. 4c). Overall, *S. pombe* cells expressing Shc1^{S.j.} showed higher FA desaturation (Fig. 4d, and Supplementary Fig. 4d, Supplementary Data 1b) and an increase in average FA chain length (Supplementary Fig. 4e, Supplementary Data 1b).

We wondered why the *S. japonicus* Shc1, even when expressed at high levels in *S. pombe*, synthesized only limited amounts of its product diplopterol. Considering that both Shc1 and the squalene epoxidase Erg1 use the same squalene substrate (Fig. 1a), we wondered if competition from native Erg1 could interfere with efficient squalene utilization by Shc1. Consistent with this possibility, Erg1 was much more abundant in *S. pombe* than in *S. japonicus* (Fig. 4e, and Supplementary Fig. 4f). We reasoned that decreasing Erg1 abundance in *S. pombe* might promote the flux of squalene towards hopanoid synthesis. To test this hypothesis, we generated *S. pombe* strains in which the endogenous *erg1* promoter was replaced by the mild constitutive *rga3* promoter[46] (*prga3:erg1*), either on its own or in combination with *ptdh1:shc1^{S.j.}-sfGFP*. Using a genetic means to downregulate Erg1 allowed us to control Erg1 levels and minimize potential off-target effects associated with terbinafine treatment[47]. Reverse-transcription qPCR analysis showed that the steady-state abundance of *erg1* mRNA in the *prga3:erg1* cells decreased ~50% of the wild-type level (Supplementary Fig. 4g).

The GC-MS analysis confirmed that the ergosterol content was indeed lower in the strains with attenuated expression of *erg1* (Fig. 4f, Supplementary Data 1c). Interestingly, we observed a considerable increase in squalene levels in the *prga3:erg1* cells. Shc1^{S.j.} expression in this mutant background led to the reduction in squalene, suggesting that the two enzymes indeed compete for the same substrate pool (Fig. 4f). Further supporting this hypothesis, diplopterol content was ~2.5 times higher in Shc1^{S.j.}-expressing *prga3:erg1* cells, as compared to the wild type with Shc1^{S.j.}. Moreover, we also detected the minor products of Shc1^{S.j.}, the hop-22(29)-ene and hop-17(21)-ene hopenes, in Shc1^{S.j.}-expressing *prga3:erg1* mutant (Fig. 4f).

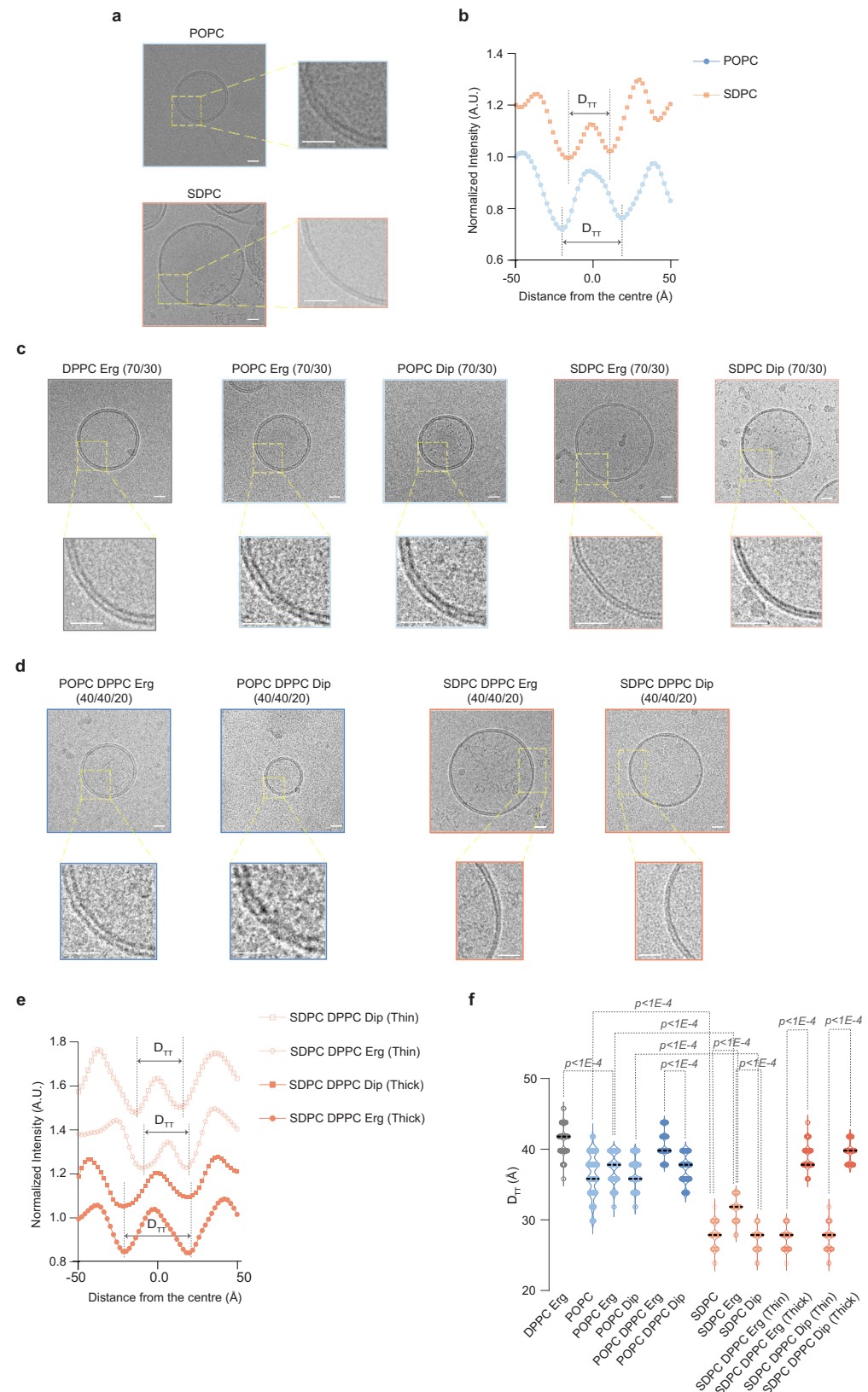

To illuminate the immediate consequences of introducing hopanoid synthesis in a eukaryote, we tested if *S. pombe* cells expressing Shc1[S.j.] differed from the wild type in several physiological scenarios, which affect membrane function. *S. pombe* growth is limited at temperatures exceeding 36 °C (Fig. 4g and Supplementary Fig. 4h). Interestingly, although the expression of Shc1[S.j.] did not affect growth within

the physiological range of this organism, it became advantageous at the higher temperature of 37 °C (Fig. 4g and Supplementary Fig. 4h). Attenuation of *erg1* expression by itself conferred a growth defect at higher temperatures (Fig. 4g and Supplementary Fig. 4h), possibly through the decrease in membrane ergosterol levels and /or accumulation of squalene[48,49] (Fig. 4f). Introducing hopanoid synthesis to

**Fig. 3 | Cryo-EM shows that asymmetrical glycerophospholipids form thinner membranes and support phase separation with either ergosterol or diplopterol. a** Representative cryo-EM images of single component LUVs made from the monounsaturated symmetrical lipid POPC (top panel) and the asymmetrical saturated SDPC (bottom panel). Right, magnified areas of respective images. **b** Representative normalized intensity profiles for POPC and SDPC liposomes. The trough-to-trough distance $D_{TT}$ is shown by arrows within dotted lines. **c** Representative cryo-EM images of two-component liposomes made from either POPC or SDPC glycerophospholipids in combination with 30 mol% ergosterol or diplopterol. The gel-like membranes composed of the symmetrical saturated lipid DPPC with ergosterol is also included. Magnified areas are shown at the bottom of respective images. **d** Representative cryo-EM images of three-component liposomes made either from 40 mol% POPC or SDPC glycerophospholipids in combination with 40 mol% gel-forming DPPC and 20 mol% ergosterol or diplopterol. Magnified areas are shown at the bottom of respective images. Note the phases of different membrane thickness in the ternary SDPC-DPPC-based membranes. **e** Representative normalized intensity profiles for thick and thin regions in phase-separated liposomes made from SDPC in the presence of DPPC and ergosterol or diplopterol. The trough-to-trough distance $D_{TT}$ is indicated by arrows. **f** The violin plot shows $D_{TT}$ values for single, two- and three-component LUVs assembled with POPC or SDPC (average ± S.D. $n = 20$ liposomes, 10 measurements for each liposome, one biological repeat). Medians are indicated. $p$ values were obtained from two-tailed unpaired parametric $t$-tests. (**a, c, d**) Scale bars represent 20 nm. Source data are provided as a Source Data file.

cells with attenuated Erg1 expression also led to the pronounced rescue of growth at higher temperatures (Fig. 4g and Supplementary Fig. 4h).

Unlike *S. japonicus*, *S. pombe* is an obligate aerobe, which can divide only a few times in fully anoxic conditions, likely due to increasing dilution of sterols, unsaturated FAs and other metabolites that require oxygen for their synthesis (Fig. 4h and ref. 8). Strikingly, as long as we supplemented the unsaturated fatty acid (UFA) Tween 80, the Shc1[S,j.]-expressing *S. pombe* could grow considerably better in the absence of oxygen (Fig. 4h). As expected, shifting *S. pombe* cells from normoxic to anoxic environment for 24 h led to a significant decrease in ergosterol levels, accompanied by an accumulation of squalene, as shown by GC-MS (Fig. 4i). The expression of Shc1[S,j.] in these conditions led to a profound increase in hopanoid synthesis, and reduced squalene levels (Fig. 4i, Supplementary Data 1d). We concluded that the optimization of SHC performance in the eukaryotic context may necessitate tuning the activity of the ergosterol biosynthetic pathway.

The introduction of Shc1 to UFA-supplemented *S. pombe* has provided us with an opportunity to inspect how the lipidome of this organism is rewired in response to oxygen limitation. Strikingly, in *S. pombe* cells shifted from normoxia to anoxia, the levels of typically minor asymmetrical saturated GPL species [26:0] and [28:0] markedly increased, concomitant with the reduction in symmetrical unsaturated GPL species [36:2] and [34:1] (Supplementary Fig. 4i, Supplementary Data 1d). Overall, *S. pombe* cells shifted to anoxia exhibited a reduction in FA unsaturation and shortening of the average FA chain length (Supplementary Fig. 4j, k, Supplementary Data 1d). Thus, even an obligate aerobe *S. pombe* might be able to tune the activity of its fatty acid synthase to produce medium-chain FAs leading to an increase in asymmetrical GPLs. Such a regulatable property could contribute to the maintenance of membrane physicochemical properties upon oxygen limitation, where FA desaturation is restricted.

We conclude that acquiring hopanoid biosynthesis through horizontal gene transfer may offer organismal advantages for exploring anoxic and warm ecological niches. Notably, integrating this new module into the metabolism of the recipient cells may require adjustments to the sterol production pathway, alongside molecular adaptations that enhance membrane functionality in the presence of both triterpenoids (Fig. 4j).

## Discussion

Unlike most eukaryotes, the fission yeast *S. japonicus* thrives in strictly anaerobic conditions. Our work suggests that the horizontal acquisition of a bacterial gene encoding a squalene-hopene cyclase[6,9] has been at the root of this physiological innovation (Fig. 1).

Shc1 appears to be deeply integrated into the lipid metabolism of *S. japonicus*, which produces both hopanoids and ergosterol under normoxic conditions and switches entirely to hopanoids for anaerobic growth. Furthermore, although sterols are typically essential to support membrane function in eukaryotic cells[50], *S. japonicus erg1Δ*

mutant cells, lacking ergosterol production altogether, are viable (Fig. 1h).

*S. japonicus* lipidome is unusually rich in asymmetrical glycerophospholipids containing long (typically C18:0) and medium fatty acyl (C10:0) chains[29]. We show that these lipids form thinner membranes, as compared to those made from the symmetrical unsaturated *S. pombe*-like glycerophospholipid POPC (Fig. 3). This observation potentially explains the shortening of transmembrane helices in a subset of proteins in *S. japonicus* as compared to *S. pombe*[29], as a strategy to reduce hydrophobic mismatch[51].

Importantly, we show that when mixed with the gel-like saturated DPPC, these asymmetrical glycerophospholipids support membrane phase separation in vitro in the presence of both the native triterpenoid ergosterol and the foreign triterpenoid diplopterol (Fig. 2d). This behavior contrasts sharply with symmetrical unsaturated glycerophospholipids DOPC and POPC, which support phase separation only in the presence of ergosterol. It is possible that accommodating bulky diplopterol molecules in the membrane requires the more relaxed lipid packing afforded by relatively compact asymmetrical glycerophospholipids[52].

Tuning the lipid composition and packing in model membranes may alter phase separation tendencies and properties of phase-separated domains. For instance, diplopterol has been shown to support phase separation in the mixtures with a synthetic sphingolipid and DOPC[53]. Notably, although diplopterol supports membrane ordering in model membranes containing asymmetrical glycerophospholipids (Fig. 2d and Fig. 3), it is a less effective order inducer than ergosterol (Fig. 2b, e–g, and Supplementary Fig. 2e–g). The situation probably differs in vivo, given that *S. japonicus* thrives in the absence of oxygen. In fact, the lack of either ergosterol or hopanoids in vivo leads to a comparable decrease in membrane order (Fig. 2h, i). Interestingly, the lack of Erg1 appears to have some deleterious effects in normoxia, suggesting that ergosterol may have specific functions in aerobic membranes. Alternatively, sterol biosynthesis could function as an oxygen sink, protecting cells from oxidative damage.

Asymmetrical C10:0- and C12:0-containing glycerophospholipids are found in other species, albeit at lower abundances[29,44]. The proportion of such lipids increases in anoxia not only in *S. japonicus* but also in *S. pombe*, as well as in budding yeast (Supplementary Fig. 1g, Supplementary Fig. 4i and refs. 54,55), suggesting a convergent lipidomic adaptation in response to oxygen deprivation. While the production of the medium-chain FAs appears to be a regulatable feature in many organisms, it might have been constitutively augmented during the evolution of *S. japonicus* following the acquisition of SHC. Indeed, engineering approaches indicate that just a few mutations can change the FAS product spectrum[56]. The ability to produce asymmetrical glycerophospholipids is advantageous not only for accommodating diplopterol but also to maintain membrane fluidity in anoxia where FA desaturation is not possible (Fig. 2 and ref. 54).

If oxygen availability is not a consideration, other GPL architectures, such as high proportion of FA desaturation, may possibly

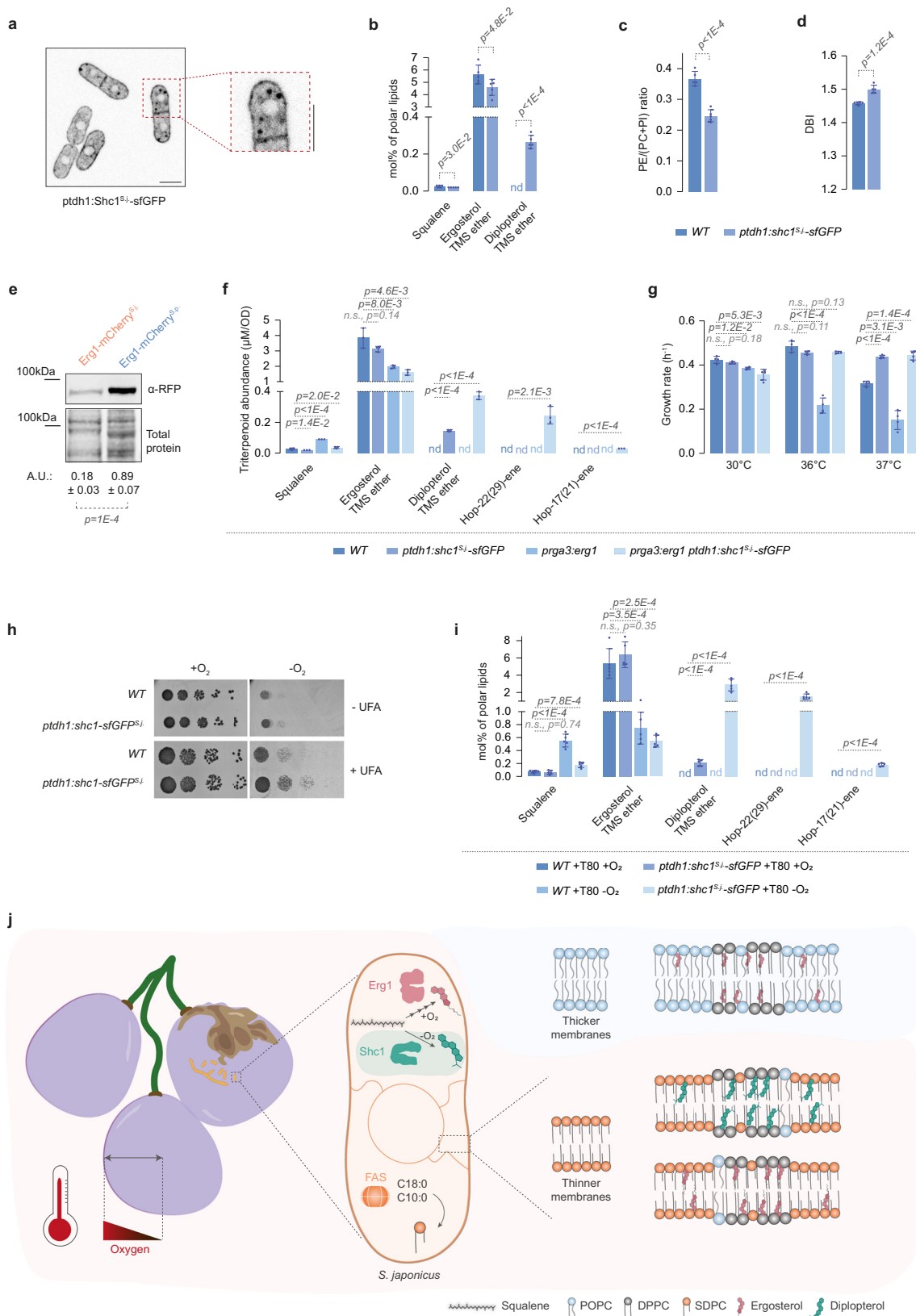

support integration of hopanoids. For instance, the membranes of *Tetrahymena pyriformis* cells have more desaturated GPLs in the presence of tetrahymanol and diplopterol as compared to ergosterol[57]. Consistent with this scenario, Shc1[S.j.]-expressing *S. pombe* cells grown in normoxia further increase FA chain unsaturation rather than synthesizing more asymmetrical glycerophospholipids (Fig. 4d).

Our data suggest that, beyond reducing oxygen dependence, the acquisition of hopanoid biosynthesis may further expand the environmental versatility of unicellular eukaryotes, including their ability to grow at elevated temperatures (Supplementary Fig. 1i, Fig. 4g, Supplementary Fig. 4h, and ref. 8). Although the molecular mechanisms underlying this effect remain to be elucidated, it is noteworthy that in

**Fig. 4 | Engineering hopanoid production in *S. pombe* offers physiological advantages but their efficient synthesis requires dampening of Erg1 levels.**
**a** Single plane spinning disk confocal image of *S. pombe* cells expressing Shc1-sfGFP under the regulation of *tdh1* promoter (*ptdh1:shc1^S.j.^-sfGFP*) grown in YES medium. A magnified image is included. Scale bars represent 5 μm. **b** Relative abundance of triterpenoids in *S. pombe wild type* (*WT*) and *ptdh1:shc1^S.j.^-sfGFP* cells. **c** PE/(PC + PI) ratio for *S. pombe WT* and *ptdh1:shc1^S.j.^-sfGFP* cells. **d** Comparison of the double-bond indexes (DBI) calculated for the four main GPL classes in *S. pombe WT* and *ptdh1:shc1^S.j.^-sfGFP* cells. **e** Western blot of Erg1 levels in *S. japonicus* and *S. pombe* cells expressing Erg1-mCherry grown in YES. Revert 700 stain was used to visualize total protein loading. Quantification is shown below the Western blot, expressed as arbitrary units. Data are represented as average ± S.D. (*n* = 3 biological repeats). Protein molecular weight standards are indicated. **f** Abundance of triterpenoids in *S. pombe* strains of indicated genotypes, as detected by GC-MS (in μM/OD). Results are represented as average ± S.D. (*n* = 3 biological repeats). **g** Comparison of growth rates of *S. pombe* strains of indicated genotypes. Cells were grown in YES at the indicated temperatures. Data are represented as average ± S.D. (*n* = 3 biological repeats). **h** Serial dilution assay of *S. pombe* strains of indicated genotypes carried out in normoxia or anoxia in YES in the absence or the presence of an unsaturated FA (UFA) supplement Tween 80. The experiment was repeated three times with similar results. **i** Relative abundance of triterpenoids in *S. pombe WT* and *ptdh1:shc1^S.j.^-sfGFP* strains grown in normoxia or in anoxia in the presence of Tween 80, as detected by GC-MS. **j** A model suggesting how the acquisition of a squalene-hopene cyclase Shc1 through horizontal transfer has led to the reorganization of *S. japonicus* lipid metabolism, allowing it to explore new ecological niches. The pictorial legend for lipids is included. **b**, **f**, **i** nd, not detected. **b–d**, **i** Data are represented as average ± S.D. (*n* = 3 biological and 2 technical repeats). **b–g**, **i** *p* values are derived from two-tailed unpaired *t*-test. Source data are provided as a Source Data file.

bacteria, hopanoids have been linked to tolerance to higher temperatures, and acidic and ethanol stresses[58–60].

Integrating SHC into the logic of ergosterol-reliant membrane lipid metabolism may require cells to co-regulate hopanoid and sterol production. Our data suggest that, at least in fission yeast, Erg1 directly competes with Shc1 for squalene, and thus, its activity must be dampened in order to allow efficient synthesis of hopanoids. Additionally, cells may need to adapt their membrane homeostasis to the presence of different triterpenoids. Accordingly, we observe large-scale changes to the cellular lipidomes both upon the loss (Fig. 1e–g, Supplementary Fig. 1c–h) and ectopic acquisition of hopanoids (Fig. 4b–d and Supplementary Fig. 4a–e) in *S. japonicus* and *S. pombe*, respectively.

Sterols are essential in most eukaryotes, and if they cannot be synthesized de novo, they must be assimilated from the environment. In such cases, the source of dietary sterols becomes critical. Potential trade-offs of sterol import could be a disruption of native membrane organization and function by foreign sterols[61]. Such considerations could drive organisms frequenting hostile environments towards different solutions, including integration of SHC enzymes. In line with this, fission yeasts cannot import sterols[62]. Similar considerations could have been at play in other lineages exhibiting SHC horizontal gene transfer. Of interest, SHCs have been acquired independently in a number of eukaryotic lineages, including several species of pathogenic fungi such as *Aspergillus fumigatus*[16], which encounter warm hypoxic environments at infection sites[63].

Horizontal gene transfer is widespread in all major eukaryotic groups and has been linked to metabolic innovations and adaption to new environments[64]. Beyond squalene hopene cyclases and related proteins, HGT events have likely contributed to other metabolic changes enabling growth under oxygen limitation. For instance, a horizontally transferred gene involved in rhodoquinone biosynthesis has facilitated respiratory chain remodeling in an anaerobic protist *Pygsuia biforma*[65], and budding yeast appears to have acquired oxygen-independent pyrimidine biosynthesis through HGT[66].

SHC is a standalone metabolic enzyme, which does not require high physical connectivity with other proteins (Fig. 4 and ref. 67). This feature, combined with the immediate adaptive benefits of hopanoids in certain environmental conditions (Fig. 4g, h and Supplementary Fig. 4h), may explain the multiple independent instances of horizontal gene transfer of SHC and SHC-related enzymes into eukaryotic lineages[17]. Importantly, our work suggests that following the initial acquisition event, the domestication of SHC may have necessitated large-scale rewiring of host metabolism with profound consequences for membrane structure and other key aspects of cellular biology. We expect that similar considerations can be at play in other cases when new metabolic traits are acquired not only through horizontal gene transfer but also through interspecific hybridization and introgression[68–70].

## Methods

### Fission yeast strains and growth conditions
The *S. japonicus* and *S. pombe* strains used in this work are listed in Supplementary Data 2. All strains were prototrophic. Standard fission yeast media and culture methods were used[71–74], with an exception of the modified minimal YNB medium (YNB containing 111 mM glucose, 14.7 mM potassium hydrogen phthalate, 15.5 mM disodium hydrogen phosphate, and the following supplements: adenine (93.75 mg/l), uracil (75 mg/l), histidine (75 mg/l) and leucine (75 mg/l). We chose a YNB-based minimal medium as an alternative to the canonically-used Edinburgh Minimal Medium (EMM)[71] since in our experimental setup, *S. japonicus* did not grow in EMM under strictly anaerobic conditions. For liquid cultures, cells were routinely grown in rich yeast extract with supplements (YES) or minimal modified YNB medium in 200 rpm shaking incubators at 30 °C, unless otherwise stated. Typically, cells were pre-cultured in YES or modified YNB over 8 h, followed by dilution to appropriate OD$_{595}$ and sub-culture overnight to reach mid-exponential phase (OD$_{595}$ 0.4–0.6) the following morning. For *S. japonicus* anaerobically grown cultures, cells were first aerobically grown overnight in either YES or modified YNB medium. The aerobic precultures were used to inoculate anaerobic precultures in the corresponding oxygen-purged media in anoxic environment inside the InvivO$_2$ 400 workstation (Baker-Ruskinn). Cells were allowed to grow until the end of exponential phase and samples from these precultures were used to inoculate the corresponding anoxic cultures. For lipidomics analyses of *S. pombe* in anoxia, cells were first aerobically grown overnight in minimal modified YNB medium supplemented with 1% of Tween 80 stock solution prepared with pure ethanol as solvent (50% v/v). Cells were collected by centrifugation at 845 × *g* and inoculated at OD$_{595}$ 0.4–0.6 in the oxygen-purged YNB medium supplemented with Tween 80 in anoxic environment inside the InvivO$_2$ 400 workstation (Baker-Ruskinn), and were allowed to grow for 8 h. Afterwards, cultures were diluted to OD$_{595}$ 0.15 and grown overnight to reach mid-exponential phase (OD$_{595}$ 0.4–0.6). *S. japonicus* and *S. pombe* mating was induced on SPA solid medium containing supplements as above at 25 °C. Spores were dissected and germinated on YES agar plates using a dissection microscope (MSM 400, Singer Instruments).

### Materials used in cell biological and biochemical experiments
D-glucose anhydrous (cat. #G/0450/60) and BD Difco™ YNB without Amino Acids (cat. #291920) were purchased from ThermoFisher. Bacteriological agar (cat. #LP0011B) was purchased from Oxoid. Sodium phosphate dibasic dihydrate (cat. #71643), potassium hydrogen phthalate (cat. #P1088), adenine hemisulfate salt (cat. #A9126), L-histidine (cat. #H8000), L-leucine (cat. #L8000), uracil (cat. #U0750), sodium sulfate (cat. #239913), DMSO (cat. #D8418), terbinafine (cat. #T8826), amphotericin B (cat. #A2942), Tween 80 (cat. #P1754), filipin (cat. #F9765), 5-α-cholestane (cat. #C8003), ergosterol

(cat. # E6510), squalene (cat. #S3626), tert-butyl methyl ether (cat. #650560), methanol (cat. #34860) and ethanol (cat. #32221) were purchased from Sigma-Aldrich. NuPage 4-12% BT gels (cat. #NP0321BOX), NuPage MOPS SDS running buffer (cat. #NP0001), NuPage transfer buffer (cat. #NP0006-1) and NuPage LDS sample buffer (cat. #NP0007) were purchased from Invitrogen. Nitrocellulose membranes 0.2 μm (cat. #1620112) were purchased from Bio-Rad. Mouse anti-RFP monoclonal antibody (cat. #6g6) was purchased from ChromoTek. Revert 700 Total Protein Stain Kit (cat. #926-11010) and IRDye 800CW Goat anti-Mouse IgG (cat. #926-32210) were purchased from LI-COR. RNeasy Plus Mini Kit (cat. #74134) and RNase-Free DNase Set (cat. #79256) were purchased from QIAGEN. Revertaid first strand cDNA synthesis kit (cat. #K1622) was purchased from Thermo Fisher. Probe Blue Mix Lo-ROX (cat. #PB20.21-01) was purchased from qPCRBIO. Micro BCA Protein Assay Kit was purchased from Thermo Fisher (cat. #23235). SPE Bulk Sorbent, primary secondary amine (PSA) (cat. #5982-8382) was purchased from Agilent. MSTFA (cat. #TS-48910) was purchased from Thermo Fisher and TSIM (cat. #MN701310.110) was purchased from Macherey-Nagel. The triterpenoid standards diplopterol (cat. #C1391.30), hop-17(21)-ene (cat. #C0789.30), hop-21(22)-ene (17β(H)) (cat. #C0699.30) and hop-22(29)-ene (cat. #C0698.30) were purchased from Chiron UK.

## Molecular genetics

All primers are shown in Supplementary Data 3. Molecular genetics manipulations were performed using PCR[75]- or plasmid[76]-based homologous recombination. To express Shc1[S.j.]-sfGFP under the *tdh1* promoter in *S. pombe*, *shc1* open reading frame (ORF) was PCR-amplified from *S. japonicus* genomic DNA and cloned into pSO1006 (pAV0749[46]) between XhoI and EcoRI enzyme sites. The resulting plasmid pSO1275 was linearized before transformation and integrated into the *ura4* locus. To build plasmid pSO1276, *rga3* promoter (1203 bp upstream of the start codon) and *kanMX6* resistance cassette plus the plasmid backbone were PCR-amplified from *S. pombe* genomic DNA and pSO257 (pKS395) respectively, and assembled using the Gibson Assembly Master Mix (New England Biolabs). Plasmid pSO1276 was then used as a template to amplify *kanMX6:prga3* flanked by 80 base pairs upstream (position -529) and downstream (position +1) of the endogenous *erg1* promoter, followed by integration into the endogenous locus, resulting in the replacement of *S. pombe erg1* endogenous promoter by a weaker *rga3* promoter. A PCR-based method was used to knock out *S. japonicus shc1* and *erg1*, as well as to tag Shc1, and *S. pombe* and *S. japonicus* Erg1 at the C-terminus using *kanR* or *natR* as selection markers. All constructs were verified by sequencing. *S. japonicus* cells were transformed by electroporation[73]. *S. pombe* transformation was performed using lithium acetate and heat shock[71]. Transformants were selected on YES agar plates containing G418 (Sigma Aldrich), nourseothricin (HKI Jena), or EMM agar plates minus uracil.

## Serial dilutions assays

*S. japonicus* and *S. pombe* cells were pre-cultured overnight in YES or modified YNB at 30 °C until early-exponential phase. Cultures were then diluted to $OD_{595}$ 0.1, and serial 10-fold dilutions were spotted on YES or YNB agar plates or the same media supplemented with different concentrations of terbinafine (Sigma-Aldrich) or amphotericin B (Sigma-Aldrich). YES plates supplemented with Tween 80 (Sigma-Aldrich) were made by adding 1% of a Tween 80 stock solution prepared with pure ethanol as solvent (50% v/v). Plates were typically incubated at 30 °C unless stated otherwise, either in the presence of oxygen or in an anoxic environment inside an InvivO2 400 workstation (Baker-Ruskinn). After 3 days, plates were scanned using an Epson Perfection V700 Photo scanner. All experiments were repeated three times with similar results, and representative experiments are shown in the corresponding figures.

## Genetic crosses between *shc1Δ* and *erg1Δ S. japonicus* mutants

We dissected 189 spores from genetic crosses between *S. japonicus shc1Δ* (SOJ5465) and *erg1Δ* (SOJ5533) strains. If the double *shc1Δ erg1Δ* mutant were viable, the expected double mutant progeny would be 47/189 (25%). We found that crosses between the single mutants led to overall high spore lethality (only 97/189 progeny were able to form colonies). Out of 97 germinated spores, we recovered 37 *erg1Δ* and 31 *shc1Δ* single mutants, and 0 double mutants. The higher-than-expected number of non-germinating spores is likely a combination of lethality of the double mutant, and the fact that single mutants themselves may exhibit some sporulation defects.

## Microscopy and image analysis

Prior to imaging, 1 ml cell culture was concentrated to 50 μl by centrifugation at $1500 \times g$ for 1 min. 2 μl cell suspension was loaded under a $22 \times 22$ mm glass coverslip (VWR, thickness: 1.5). Fluorescence images in Fig. 1b, Fig. 4a and Supplementary Fig. 4f were acquired using Yokogawa CSU-X1 spinning disk confocal system with Eclipse Ti-E Inverted microscope with Nikon CFI Plan Apo Lambda 100× Oil N.A. = 1.45 oil objective, 600 series SS 488 nm SS 561 nm lasers, and Andor iXon Ultra U3-888-BV monochrome EMCCD camera controlled by Andor IQ3. Single plane images with inverted LUT (look-up-table) are shown. Images shown in Fig. 1h and Supplementary Fig. 1a were captured using a Zeiss Axio Observer Z1 fluorescence microscope fitted with α Plan-FLUAR 100×/1.45 NA oil objective lens (Carl Zeiss) and the Orca-Flash4.0 C11440 camera (Hamamatsu). Images were taken at the medial focal plane of cells.

Filipin staining of sterols was performed by adding the drug at a final concentration of 5 μg/ml from a DMSO stock to the cell liquid cultures in YES medium. Cells were observed immediately upon drug addition.

Image analysis and quantification were performed using Fiji[77]. Within the same experiment, images are directly comparable as they are adjusted to equal brightness and contrast levels. Measurements of cellular length and width were performed on bright-field images acquired with the Zeiss epifluorescence microscope.

## Cell growth and colony forming unit (CFU) assays

For growth rate experiments, *S. japonicus* and *S. pombe* cells were grown overnight in YES or modified YNB either at 30 °C, 36 °C, or 37 °C until early-exponential phase. Cultures were then diluted to $OD_{595}$ 0.1-0.15 with the same medium and loaded into a 96-well plate. Growth was measured every 60 min at the corresponding temperature using VICTOR Nivo multimode plate reader (PerkinElmer). Growth rates and Tmid were calculated using the Growthcurver R package[78]. Experiments were repeated at least three times from cultures grown on separate occasions.

CFU measurements were performed as described in ref. 79 with minor modifications. Briefly, *S. japonicus* cells were pre-cultured overnight in YES or modified YNB at 30 °C until stationary phase. Cultures were normalized to $OD_{595}$ 1 and three decimal dilutions ($10^{-1}$, $10^{-2}$, $10^{-3}$) of each were prepared. 100 μl of the $10^{-3}$ dilution were further diluted with 500 μl of the appropriate media, and 100 μl of the resultant samples were plated in quadruplicates on YES or modified YNB agar plates. After 2 days (YES plates) or 3 days (modified YNB plates) plates were scanned using an Epson Perfection V700 Photo scanner. The colony numbers and size were measured using Fiji[77].

## Western blotting of Erg1-mCherry

*S. japonicus* and *S. pombe* cell cultures were grown in YES to mid-exponential phase. 5 ODs were pelleted for 1 min at $2103 \times g$ and supernatant was removed. Cells were resuspended in 1 ml ice-cold dH2O and transferred to 1.5 ml Eppendorf tubes. Cells were washed and snap-frozen in liquid nitrogen. Western blotting experiments were performed as previously described in ref. 80 with minor modifications.

Briefly, 110 µl of ice-cold TCA was added to the cell pellet to allow protein precipitation for 1 h on ice. Lysates were centrifuged for 10 min at 18213 × g at 4 °C. The supernatant was removed and the pellet was washed in 1 ml of ice-cold acetone once. After removing the supernatant, pellets were dried in a speed-vac (Eppendorf) for 2 min at room temperature. 300 µl of boiling buffer (50 mM Tris pH 8.0, 1 mM EDTA, 1% SDS) were added to the pellets and transferred to chilled screw-cap tubes with zirconium beads. Cells were disrupted using a MP Biomedicals cell disruptor for 2 × 15 s at 4 °C and 6.5 m/s with 2 min cooling down at 4 °C in between. Lysates were then separated from the beads using a hot needle and spun down for 3 min at 526 × g at 4 °C. Debris were cleared by centrifugation of lysates at 1383 × g for 4 min at room temperature. Lysates were boiled at 95 °C for 5 min. 100 µl of 4 × 10% β-mercaptoethanol sample buffer was added to the lysates, which were heated at 65 °C for 10 min. 20 µl of lysate was loaded per lane on a NU-PAGE 4–12% Bis-Tris gel (Invitrogen) and run at 130 V for 2 h. Proteins were transferred at 100 V for 1 h. For Erg1-mCherry detection nitrocellulose 0.2 µm membranes (Bio-Rad) were blocked for 1 h with TBST buffer containing 5% skimmed milk and then incubated for 1 h with mouse α-RFP antibody (ChromoTek) at 1:200 dilution. Membranes were washed with TBST and incubated for 1 h in TBST buffer with IRDye800 conjugated α-mouse antibody (LI-COR Biosciences) at 1:10000 dilution. Total protein levels were detected using the LI-COR Revert 700 Total Protein stain kit (LI-COR Biosciences). Proteins were detected using the ChemiDoc MP imaging system (Bio-Rad). Samples were collected as at least three biological replicates from cultures grown on separate occasions. Quantification was performed using Fiji[77].

### Reverse transcription and real-time quantitative PCR (RT-qPCR)

Reverse transcription and qPCR were performed as previously described in ref. [29]. Briefly, *S. pombe* cultures were grown in YES to mid-exponential phase and 10 ODs of cells were collected and subjected to total RNA extractions using a QIAGEN RNeasy Plus Mini Kit. Reverse transcription of RNA was performed at 50 °C for 1 h in a total reaction volume of 20 µl containing 1 µg RNA and 0.5 µg oligo(dT) using the Transcriptor First Strand cDNA Synthesis Kit (Roche). qPCRBIO Probe Blue Mix Lo-ROX was used for the real-time qPCR (RT-qPCR). The RT-qPCR was performed on a LightCycler 96 Instrument (Roche Diagnostics) in three biological and two technical repeats. RT-qPCR signal was normalized to actin (*act1*) expression levels.

### Triterpenoid detection by GC-MS

*S. japonicus* and *S. pombe* cell cultures were grown in modified YNB to mid-exponential phase and 10 ODs were harvested via filtration, then snap frozen in liquid nitrogen. For triterpenoid detection, a total of five replicates were collected per condition (three biological repeats, two technical repeats). Metabolite extraction for triterpenoid detection was performed as described in refs. [6,81] with minor modifications. Before processing, cell pellets were lyophilized overnight using a freeze dryer lyophilizer (Labconco). For cell lysis, each freeze-dried pellet was mixed with 1 ml of 2 M NaOH, transferred to glass vials and heated for 1 h at 70 °C in a water bath, with vortexing at 15 min intervals. After saponification, suspensions were allowed to cool to room temperature and divided into two microcentrifuge tubes (2 × 500 µl). 650 µl of distilled methyl-tert-butylether (MtBE, Sigma-Aldrich, HPLC grade) and 100 µl of internal standard solution (5α-cholestane in MtBE, 10 µg/ml) were added to each sample. Mixtures were vortexed for 1 min and centrifuged at 9000 × g for 5 min. After centrifugation, the organic upper layer (~550 µl) was transferred into a new microcentrifuge tube containing 40 ± 2 mg of a mixture (7:1) of anhydrous sodium sulfate (Sigma-Aldrich) and primary secondary amine (PSA, Agilent Technologies) (dispersive solid phase). Lysed cells were subjected to a second round of metabolite extraction by adding another

750 µl of MtBE, shaking for 1 min and centrifuging at 9000 × g for 5 min. The organic upper layer (~650 µl) was transferred to the tube containing the dispersive solid phase and combined with the previous organic extract. The mixtures were shaken and centrifuged again, and 1 ml of the purified upper layer was transferred to an amber glass vial (Agilent Technologies).

Samples were evaporated to dryness under a stream of nitrogen at room temperature. For each pair of vials, one residue was resuspended in 700 µl of MtBE and 50 µl of silylation reagent mixture MSTFA/TSIM (9:1) and the other residue was resuspended in 750 µl of MtBE. Samples were vortexed for 10 s and incubated for at least 30 min at room temperature.

Triterpenoid analysis was performed by GC-MS using an Agilent 7890B-5977A system. Splitless 1 µl injection (injection temperature 250 °C) onto a 40 m × 0.25 mm VF-5ms + EZ Guard column (Agilent J&W) was used with helium as the carrier gas, in electron impact (EI) ionization mode. The initial oven temperature was 55 °C (1 min), followed by temperature gradients to 260 °C at 50 °C/min, and then to 320 °C at 4 °C/min (held for 4 min). Mass spectra were acquired at 70 eV in the range from 50 m/z to 600 m/z. Triterpenoid identification was performed by comparison to retention time and fragment ion pattern of authentic standards using MassHunter Workstation software (B.06.00 SP01, Agilent Technologies) and confirmed by comparison to deconvoluted mass spectra of those in the NIST Mass Spectral Library software (NIST 23, software version 3.0). The following standards were used: diplopterol, ergosterol, hop-17(21)-ene, hop-21(22)-ene (17β(H)), hop-22(29)-ene and squalene. A 6-point calibration curve of the GC-MS system with standard mixtures was used for relative quantification of triterpenoid compounds. GC-MS quantification details for external and internal standards are found in Supplementary Data 1e.

### ESI-MS-based lipidomic analysis

Lipid standards were from Avanti Polar Lipids (Alabaster, AL, USA). Solvents for extraction and MS analyses were liquid chromatography grade (Merck, Darmstadt, Germany) and Optima LC-MS grade (Thermo Fisher Scientific, Waltham, MA, USA), as applicable. All other chemicals were the best available grade purchased from Sigma-Aldrich or Thermo Fisher Scientific.

10 ODs of exponentially growing yeast cell cultures in modified YNB were pelleted and frozen in liquid nitrogen. For lipidomics analysis, a total of five replicates were collected per condition (three biological repeats, two technical repeats). Pellets were disrupted in water using a bullet blender homogenizer (Bullet Blender Gold, Next Advance, Inc) in the presence of zirconium oxide beads (0.5 mm) at speed 8 for 3 min at 4 °C to achieve a homogenate concentration of 20 OD/ml. Protein concentration of cell homogenates was determined using the BCA Protein Assay Kit (Thermo Fisher Scientific). A portion of the homogenate (30 µl of ~600 µl total volume) was immediately subjected to a simple one-phase methanolic (MeOH) lipid extraction[45]. First, the homogenate was sonicated in 0.5 ml MeOH containing 2 µg di20:0-PC (as extraction standard) and 0.001% butylated hydroxytoluene (as antioxidant) in a bath sonicator for 5 min, then shaken for 5 min and centrifuged at 10,000 × g for 5 min. The supernatant was transferred into a new Eppendorf tube and stored at −20 °C until MS measurement. Electrospray ionization mass spectrometry (ESI-MS)-based lipidomic analyses were performed on an Orbitrap Fusion Lumos instrument (Thermo Fisher Scientific) equipped with a robotic nanoflow ion source TriVersa NanoMate (Advion BioSciences) using chips with spraying nozzles having a diameter of 5.5 µm. The ion source was controlled by Chipsoft 8.3.1 software. The ionization voltages were +1.3 kV and −1.9 kV in positive and negative mode, respectively, and the back-pressure was set at 1 psi in both modes. The temperature of the ion transfer capillary was 260 °C. Acquisitions were performed at the mass range

of 350–1200 m/z at the mass resolution Rm/z 200 = 240,000 in full scan mode. The lipid classes phosphatidylcholine (PC), lysophosphatidylcholine (LPC), diacylglycerol (DG), triacylglycerol (TG) and ergosteryl ester (EE) were detected and quantified using the positive ion mode, while phosphatidylethanolamine (PE), mono- and dimethylphosphatidylethanolamine (MMPE and DMPE), phosphatidylinositol (PI), phosphatidylserine (PS), their lyso derivatives LPE, LPI, LPS, phosphatidic acid (PA), phosphatidylglycerol (PG), cardiolipin (CL), ceramide (Cer), inositolphosphoceramide (IPC) and mannosylinositolphosphoceramide (MIPC) were detected and quantified using the negative ion mode.

For quantification, 10 µl of lipid extracts (corresponding to 2–3 µg protein) were diluted with 110 µl infusion solvent mixture (chloroform:methanol:iso-propanol 1:2:1, by vol.) containing an internal standard mix (35 pmol PC(15:0/18:1-d7), 24 pmol PE(15:0/18:1-d7), 20 pmol PI(15:0/18:1-d7), 11 pmol PS(15:0/18:1-d7), 4 pmol PG(15:0/18:1-d7), 4 pmol PA(15:0/18:1-d7), 2 pmol CL(tetra14:1), 5 pmol Cer(t18:0/16:0), 5 pmol DG(15:0/18:1-d7), 7 pmol TG(15:0/18:1-d7/15:0), and 9 pmol CE(18:1) (Supplementary data 1f). Next, the mixture was halved, and 5% dimethylformamide (additive for the negative ion mode) or 3 mM ammonium chloride (additive for the positive ion mode) were added to the split sample halves. 10 µl solution was infused and data were acquired for 1.2 min.

Lipids were identified by the LipidXplorer software[82] by matching the m/z values of their monoisotopic peaks to the corresponding elemental composition constraints. The mass tolerance was 3 ppm. Data files generated by LipidXplorer queries were further processed by custom Excel macros. Lipid classes and species were annotated according to the shorthand classification systems for lipids[83] at the level of sum formulas. In sum formulas, e.g., PC (34:1), the total numbers of carbons followed by double bonds for all chains are indicated. For sphingolipids, the sum formula, e.g., Cer (44:0:4), specifies first the total number of carbons in the long chain base and FA moiety, then the sum of double bonds in the long chain base and the FA moiety followed by the sum of hydroxyl groups in the long chain base and the FA moiety.

Lipidomics data are expressed as mol% of polar lipids; polar lipids include all measured lipids except DG, TG, EE, and sterols. Double bond index (DBI), average chain length, and average lipid species profile was calculated for the sum of major GPLs (PC, PI, PE, and PS). DBI was calculated as $\Sigma(db \times [GPLi])/\Sigma[GPLi]$, where db is the total number of double bonds in fatty acyls in a given GPL species, and the square bracket indicates mol% of GPLs. Average chain length was calculated as $\Sigma(C \times [GPLi])/\Sigma[GPLi]$, where C is the total number of carbons in fatty acyls in a given GPL, and the square bracket indicates mol % of GPLs. ESI-MS quantification details for internal standards are found in Supplementary Data 1f.

## Materials used in biophysical experiments
DMSO (cat. #D8418), D-glucose (cat. #G/0450/60), DPH (cat. #D208000), EDTA (cat. #E5134), ergosterol (cat. #E6510), HEPES (cat. #H3375), NaCl (cat. #S7653), Na$_2$HPO$_4$.2H$_2$O (cat. #71643), NaH$_2$PO$_4$.H$_2$O (cat. #71507) were purchased from Sigma-Aldrich. Sucrose (cat. #F492423) was purchased from Fluorochem. FAST DiI (cat. #D7756) was purchased from Thermo Fisher Scientific and C-laurdan (cat.#7273) was purchased from Tocris Bioscience. Synthetic glycerophospholipids including POPC (850457C), DPPC (850355C) and DOPC (850375C) were purchased from Avanti Polar Lipids. 18:0/10:0 PC (SDPC) was custom-synthesized by Avanti Polar Lipids. 17Beta(H),21Beta(H)-22-Hydroxyhopane (cat #C1391.30) commonly known as diplopterol was purchased from Chiron UK. Spectroscopic grade solvents such as methanol (cat. #154903) and chloroform (cat. #366919) used for lipid stock preparation were purchased from Sigma-Aldrich. Milli-Q water was used throughout. C-laurdan stock solution (100 µM) was prepared in DMSO.

## Preparation of giant unilamellar vesicles (GUVs)
GUVs were prepared by electroformation using the Nanion Vesicle Prep Pro (Nanion Technologies, Munich, Germany) as described previously[84,85], with minor modifications. Briefly, lipids (200 nmol for two-component liposomes or 300 nmol for three-component liposomes) along with 0.2 mol% FAST DiI probe were dried onto the conductive side of the ITO-coated slide. The glass slide was dried in vacuum for ~2.5 h. A medium O-ring was coated with vacuum grease and placed around the dried lipid film. 270 µl of 250 mM sucrose was added inside the O-ring, and the second conductive slide was placed on the top, making a sandwich. The standard protocol for vesicle preparation of 120 min was used with a rise and fall of 3 min. The amplitude was set at 10 Hz, voltage was 3 V, and the temperature was 55 °C. For C-laurdan labeling, the probe was added to electroformed GUVs from a 100 µM stock solution in DMSO such that the final lipid-to-probe ratio was 300:1 (mol/mol). GUVs were stored at 4 °C and used for imaging measurements on the following day. For spinning disk confocal microscopy measurements, 20 µl of GUV solution was slowly added to a microscope chamber (ibidi, Gräfelfing, Germany) filled with ~400 µl of 250 mM glucose solution for 2 h. This allowed vesicles to settle onto the bottom of the chamber before imaging measurements were carried out.

## Preparation of large unilamellar vesicles (LUVs)
LUVs were prepared as described previously with slight modifications[86]. Specifically, for fluorescence anisotropy measurements, 300 nmol lipids and 3 nmol DPH (lipid-to-probe ratio 300:1) were mixed and dried using a nitrogen stream while being warmed gently at 37 °C. For cryo-EM measurements, liposomes were prepared using 600 nmol of lipids. The lipid samples were dried in vacuum for 3 h, followed by hydration at 60 °C for 1 h in 1 ml of buffer A (10 mM sodium phosphate, 150 mM NaCl, pH 7.4) for fluorescence anisotropy measurements or buffer B (10 mM HEPES, 150 mM NaCl, pH 7.4) for cryo-EM. Samples were vortexed for 1 min to form homogeneous multilamellar vesicles (MLVs). LUVs were prepared by extrusion using Avestin Liposofast Extruder (Ottawa, Canada) as previously described[87]. Briefly, MLVs were freeze-thawed five times using liquid nitrogen and extruded through polycarbonate filters (pore diameter of 100 nm) mounted on extruder fitted with Hamilton syringes (Hamilton Company, Reno, NV). The samples were subjected to uneven passes (21 passes for all liposomes except the gel-like liposomes with DPPC and ergosterol, for which 33 passes were used) on a 55 °C hot plate. LUV sizes were measured using dynamic light scattering (Malvern Zetasizer Nano ZS). For anisotropy experiments, samples were kept overnight at 25 °C before measurements. For cryo-EM experiments, LUVs were stored at 4 °C overnight before preparing samples.

## Water permeability measurements
For water permeability assays, unlabeled single or two-component MLVs were prepared in Tris–HCl 5 mM, pH 7.0, 100 mM sucrose (isotonic buffer)[88]. Isotonic MLV suspensions were diluted by adding 0.12 ml of MLVs to 2 ml of hypotonic buffer (Tris–HCl 5 mM, pH 7.0) equivalent to ~16 times dilution. The time-dependent reduction in absorbance at 550 nm (Supplementary Fig. 2c) due to loss in turbidity was monitored for 6 min, and used to calculate permeability coefficients (Supplementary Fig. 2d). Absorbance was measured using V-560 spectrophotometer (Jasco, West Yorkshire, UK) at room temperature (~25 °C).

## Fluorescence anisotropy measurements in liposomes
Steady-state fluorescence anisotropy measurements were performed at 25 °C on a FP-8500 spectrofluorometer (Jasco, West Yorkshire, UK) with in-built excitation and emission polarizers. For monitoring DPH fluorescence, the excitation wavelength was set at 358 nm and emission was monitored at 430 nm. Quartz cuvettes with a path length of

1 cm were used. Excitation and emission slits with bandpass of 5 and 10 nm were used for all measurements. Fluorescence was monitored with a 30 s interval between successive openings of the excitation shutter to reverse any photoisomerization of DPH[89]. Anisotropy values were obtained simultaneously along with the measurements according to the equation[90]:

$$r = (I_{VV} - G \times I_{VH})/(I_{VV} + 2G \times I_{VH}) \qquad (1)$$

where $I_{VV}$ and $I_{VH}$ are the fluorescence intensities measured with the excitation polarizer oriented vertically and the emission polarizer oriented vertically and horizontally oriented, respectively. G is the grating factor and is the ratio of the efficiencies of the detection system for vertically and horizontally polarized light, and is equal to $I_{HV}/I_{HH}$. G factor was measured before sample measurements and was always ~1.

### Visualizing phase separation in model membranes
Images of FAST DiI-labeled GUVs were obtained using Yokogawa CSU-X1 spinning disk confocal system mounted on the Eclipse Ti-E Inverted microscope with Nikon CFI Plan Apo Lambda 100X Oil N.A. = 1.45 oil objective, 600 series SS 488 nm, SS 561 nm lasers and Andor iXon Ultra U3-888-BV monochrome EMCCD camera. Excitation laser of 561 nm was used. Z-stacks were acquired with step size of 0.6 μm for 12 steps. Imaging was performed at 25 °C. Image processing and quantifications were performed in Fiji[77]. Fluorescence images are shown with inverted LUT (look-up table) (Fig. 2c, d, and Supplementary Fig. 2b).

### Confocal spectral imaging for estimating membrane order
Spectral imaging of GUVs was performed on a Zeiss LSM 880 confocal microscope equipped with a 32-channel GaAsP detector array, as described previously[91]. Excitation wavelength of 405 nm was used for fluorescence excitation of C-laurdan while the lambda detection range was set between 415 and 691 nm. The intervals between the individual detection channels were set to 8.9 nm which allowed the simultaneous coverage of the whole spectrum with 32 detection channels. Images of phase separated GUVs were acquired with 1024 × 1024 resolution. Lo and Ld membrane regions were identified based on the color in a composite image obtained upon merging images corresponding to the blue-shifted and red-shifted wavelengths. The .czi format images were analyzed using a custom GP plug-in compatible with Fiji/ImageJ[91]. The fluorescence intensities obtained after processing the confocal images were used to calculate GP (generalized polarization) according to the equation:

$$GP = (I_B - I_R)/(I_B + I_R) \qquad (2)$$

where $I_B$ and $I_R$ are the fluorescence intensities corresponding to the blue-shifted wavelength (442 nm) and the red-shifted wavelength (496 nm) for C-laurdan (Fig. 2b–g and Supplementary Fig. 2f–h). Confocal imaging measurements were carried out at 37 °C.

### Live confocal microscopy-based measurements of membrane order
Cells were pre-cultured overnight in modified YNB at 30 °C and diluted to $OD_{595}$ 0.1–0.15 in the same medium and allowed to grow to mid-exponential phase of $OD_{595}$ 0.4–0.6. Harvested cells were labeled with 5 μM C-laurdan and incubated at 25 °C for 20 min. Following incubation, cells were resuspended in fresh medium and mounted on a glass slide. Imaging was carried out on a Nikon AXR inverted confocal microscope with NSPARC1 enabled with 2 GaAsP detectors. A 405 nm laser was used for excitation of C-laurdan. The detection wavelengths chosen were 420–460 nm and 470–510 nm. Acquisitions were performed immediately after sample preparation.

All measurements were done at room temperature (~25 °C). The.nd2 format images were analyzed using a custom GP plug-in compatible with Fiji/ImageJ[92].

### Cryo-EM grid preparation, imaging, and data processing
A thin carbon film was deposited onto a mica sheet using EMITECH K950X, and transferred onto Quantifoil (200 Cu mesh, R2/2) grids. The carbon film grids were then glow discharged (EMITECH K100X, 25 mA, 30 s) in an amylamine atmosphere. 4 μl of sample was pipetted to the grid in the environmental chamber of a Vitrobot Mark IV (FEI/Thermo) at 4 °C and 95-100% humidity. The grid was blotted for 1.5 s before plunging into liquid ethane kept at liquid nitrogen temperature. The grids were imaged on a Talos Arctica microscope (FEI/Thermo) at 200 kV using EPU software (v 2.11). Movies were recorded on a Falcon III camera in linear mode with a total dose of 48 electrons per Å2 fractionated over 10 frames (dose rate 24 e⁻/Å2/s) with a 1.99 Å pixel size and a nominal defocus of −2 μm. All movies were imported into Relion (v 4.0.0)[93], followed by Relion's own motion correction and CTF estimation (CTFFIND, v 4.1.13)[94]. The images within the defocus range from −1.5 μm to −2.1 μm were selected, then converted to 16-bit tiff format for further measurements. Subsequent analysis of images was performed using Fiji/Image. A 2-point Gaussian filter was applied for clear distinction of dips in the intensity profile before computation of $D_{TT}$[42] from the images.

### Statistics and reproducibility
The statistical details of experiments, including the number of biological and technical replicates and the dispersion and precision measures can be found in Figure Legends, Supplementary Fig. Legends and Methods. All data were analyzed using unpaired $t$-test statistical analysis, unless indicated otherwise. All plots were generated using GraphPad Prism 10.

No data were excluded in the cell biological and physiological experiments. In the measurements of triterpenoid abundance, linear regression analyses between sample $OD_{595}$ and lipid content were performed for each experimental group. Samples showing anomalous lipid levels relative to the sample amount were excluded as likely sample preparation artifacts.

### Reporting summary
Further information on research design is available in the Nature Portfolio Reporting Summary linked to this article.

## Data availability
All data presented in graphs and uncropped scans of all blots generated in this study are included in the Source Data file. Lipidomics data are provided in Supplementary Data 1. All microscopy, cryo-EM, and Western blotting data have been deposited in the Figshare database under accession code https://doi.org/10.6084/m9.figshare.c.7668101. Raw lipidomics data have been deposited in the Zenodo database under accession code https://doi.org/10.5281/zenodo.15017552. Source data are provided with this paper.

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

## Acknowledgements

We are grateful to the Oliferenko lab for discussions and Eugene Makeyev for suggestions on the manuscript. Many thanks to Fred Betts Thompson for media preparations, Nikita Komarov for helping with *shc1Δ* mutant construction, Laura Masino for advice on spectroscopy, and to the Crick Metabolomics, Light Microscopy and Structural Biology STPs for invaluable training and assistance. We thank Dylan Herzog, Vincenzo Infante and Stefania Marcotti at the King's Nikon Imaging Centre for their help in setting up live cell membrane order measurements. Elisa Gomez-Gil has been supported through a long-term EMBO postdoctoral fellowship (ALTF 712-2022, E.G.G.) and the UKRI guarantee of a MSCA postdoctoral fellowship (EP/Y024702/1, E.G.G) We thank the Single Cell Omics Advanced Core Facility staff of the HCEMM and Biological Research Center for help with their resources and their support. HCEMM has received funding from the EU's Horizon 2020 research and innovation program under grant agreement No. 739593 and KIM NKFIA 2022–2.1.1-NL–2022-00005. The work in the Rosenthal lab is supported through grant CC2106 from the Francis Crick Institute. This work was supported by the Francis Crick Institute, which receives its core funding from Cancer Research UK (CC0102), the UK Medical Research Council (CC0102), and the Wellcome Trust (CC0102), and the Wellcome Trust Senior Investigator Award (103741/Z/14/Z) and Wellcome Trust Investigator Award in Science (220790/Z/20/Z) to S.O.

## Author contributions

B.D.R. conceived and performed biophysical experiments; analyzed data; and co-wrote the manuscript. E.G.G. conceived and performed cell biological and biochemical experiments; generated strains; analyzed data; and co-wrote the manuscript. M.P. and G.B. designed, performed, and interpreted all ESI-MS lipidomics experiments and edited the manuscript. V.N. and J.I.M. designed, performed, and interpreted GC-MS experiments and edited the manuscript. Q.C. and P.B.R. designed, performed, and interpreted cryo-EM experiments and edited the manuscript. S.O. conceived and interpreted experiments and co-wrote the manuscript. This research was funded in whole, or in part, by the Wellcome Trust (103741/Z/14/Z; 220790/Z/20/Z to S.O.). For the purpose of Open Access, the author has applied a CC-BY public copyright licence to any Author Accepted Manuscript version arising from this submission.

## Competing interests

The authors declare no competing interests.
