## [Transparent Peer Review file · Nature Communications]

Horizontal acquisition of prokaryotic hopanoid biosynthesis reorganizes membrane physiology driving lifestyle innovation in a eukaryote

Corresponding Author: Professor Snezhana Oliferenko

Version 0:

Reviewer comments:

Reviewer #1

(Remarks to the Author)

This study investigates the impact of horizontal gene transfer on lipid metabolism and cell physiology. The authors explore the role of squalene-hopene cyclase (Shc1) in hopanoid biosynthesis in *Schizosaccharomyces japonicus* and assess its functional integration when introduced into *S. pombe*, a related species that lacks hopanoids. By comparing lipid compositions and growth dynamics in both species, the authors demonstrate how Shc1 enables *S. pombe* to withstand anoxia and high temperatures. They attribute these resilience traits to changes in membrane properties and further validate findings using synthetic membrane models. This work reveals how a single gene transfer can induce physiological changes in growth and membrane composition, with potential implications for cell membrane biology and evolution.

However, the manuscript requires further clarification of the physiological and biophysical relevance of several key findings. Specifically, the data on diplopterol's influence on membrane order and phase behavior lacks sufficient discussion of previous studies and the significance of phase separation. Additionally, certain experiments involving membrane property probes would benefit from improved controls and interpretation. Seeking input from a membrane biophysicist would enhance the framing of these results. Below are detailed comments on major and minor points.

<h3>Major Points</h3>

- **Lipidome Comparison:** The manuscript characterizes membrane lipid compositions in *S. japonicus* and *S. pombe* when either hopanoids, sterols, or both are present. However, it lacks a clear comparative analysis of how these lipid trends converge or diverge between the two species, particularly in the context of the synthetic membrane system used.
- **Live Cell Membrane Measurements:** This study would provide much deeper insight if the predicted impact of hopanoids or sterols on membrane properties were substantiated by direct measurements in live cells or membrane extracts, rather than inferred solely from model membrane systems.
- **Gene Regulation Analysis:** Introducing Shc1 likely alters gene regulation of the whole cell, which could contribute to observed metabolite changes. A more comprehensive analysis or prediction of pathway regulation is necessary to elucidate these regulatory shifts.

• **Phase Separation and Physiological Relevance:** The section exploring diplopterol's role in promoting phase separation in giant unilamellar vesicles (GUVs) composed of yeast-like lipids is promising but lacks critical context. The manuscript does not reference foundational studies, such as Saenz et al. (PNAS, 2012: <https://www.pnas.org/doi/10.1073/pnas.1212141109>), which first demonstrated diplopterol's ability to induce lo/ld phase separation and form an ordered phase with saturated lipids. The authors should discuss the biological significance of membrane phase separation, drawing on insights from studies involving GUVs, GPMVs, and observations in yeast and mammalian cells.

• **FASTDiI Staining and Membrane Phase Interpretation:** "When combined with 30 mol% of either ergosterol or diplopterol, all three glycerophospholipids formed disordered membranes, as indicated by the labeling of GUVs with FASTDiI." Staining with fastDiI is not a diagnostic indicator of whether a membrane is lo or ld, unless you are staining a phase separated vesicle where both lo and ld phases are present. The dye will preferentially partition in the presence of two phases, but it can still partition into mono-phase vesicles that are disordered. If referring to single-phase GUVs, alternative measurements such as DPH anisotropy or laurdan GP should be used to support these conclusions.

• **Laurdan GP and DPH Anisotropy in Phase-Separated Systems:** Laurdan GP and DPH anisotropy values in three-component GUVs (Fig. 2d-f) do not provide reliable insights here, as fluorescence probes can partition differently between coexisting phases. These values will not report an 'average' of membrane properties if they are phase separated, and should be removed. To validate claims, the authors should measure laurdan GP or DPH anisotropy with microscopy to discern phase-specific differences.

<h3>Minor Points</h3>

• **Triterpenoid Pathway Disruption and Membrane Integrity:** The manuscript should discuss the potential impact of squalene accumulation on membrane integrity, especially considering the reduced growth rate of the constitutive *prga3 S. pombe* strain at high temperatures. I recall there is at least one published study that has demonstrated squalene accumulation can destabilize membranes in yeast (for example: <https://pubmed.ncbi.nlm.nih.gov/22342273/>).

• **Use of Constitutive Strain in *S. pombe*:** The authors chose a constitutive *S. pombe* strain rather than using terbinafine to modulate ergosterol levels, as in *S. japonicus*. An explanation for this approach would enhance the reader's understanding.

• **Anaerobic Growth of *S. japonicus*:** The morphological differences observed in *S. japonicus* under anaerobic conditions without ergosterol supplementation should be discussed, as this could provide insights into hopanoid functionality. Please discuss how *S. japonicus* morphology in anaerobic conditions differ without access to ergosterol.

• **Explanation of SoamD Score:** The results section should briefly explain the SoamD score and its application, as this metric is not widely known among readers.

• **Unsupported Claims:** The sentence beginning with "For instance, diplopterol ..." describes phase separation with diplopterol, sphingolipid, and DOPC, however, this was not supported by the corresponding citation.

• **Integration of SHC and Fatty Acid Desaturation:** The claim that fatty acid desaturation facilitates SHC integration into membranes lacks citation. This claim was not supported by the corresponding citation. Supporting evidence for this assertion is needed to substantiate it. Please double check the accuracy of all citations.

• Please include line numbering in future revisions to help reviewers describe where in the manuscript they have concerns.

Reviewer #2

(Remarks to the Author)

This beautiful study explores how horizontal transfer of a bacterial squalene-hopene synthase gene, SHC1, into the genome of *S. japonicus* allowed this yeast to adapt to growth in complete anaerobiosis by avoiding the need to produce the normal sterols that require oxygen for their synthesis. The authors show how hopanoids can replace sterols both in vivo as well as

in vitro (artificial model membranes) on the condition that there is also an increase in asymmetrical glycerophospholipids in the membranes. Lastly, they mimic the horizontal gene transfer event by introducing SHC1 into the genome of *S. pombe*, a yeast related to *S. japonicus* but lacking its capacity to grow in anoxia. This last set of experiments confirms that SHC1 expression in *S. pombe* indeed confers better growth in anoxia, although supplementation with unsaturated fatty acids was still necessary. In addition, expression of SHC1 also bestowed *S. pombe* with a tolerance to higher temperatures, mimicking the temperature tolerance of *S. japonicus*.

Together, this study unveils the molecular details of how horizontal gene transfer bestows important new capacities onto the receiving organism, but at the meantime also requires further adaptation and tuning of the receiving organism's general physiology. It is exactly this last part that makes the study innovative and interesting. Plus, apart from helping to understand the evolutionary processes related to horizontal gene transfer, the study also adds to our understanding of anaerobic growth and lipids in general.

The paper is beautifully written and, despite its multi-disciplinary nature easy to follow even for non-experts. That said, since I am not an expert in lipid metabolism, I cannot comment in detail on the scientific rigor of the results regarding lipid composition in general, and the artificial membranes in particular. Instead, I focus on the molecular-evolutionary part. I only have some minor questions and suggestions.

1. In the introduction, it could be interesting to mention whether there are other known cases of organisms adapting to anoxia, and how they managed (are there other cases of SHC, as well as other cases of HGT that required further tuning of the receiving host organism).
2. It could be interesting to mention in the intro and/or discussion that the general principles found here may not only apply to horizontal gene transfer, but also to interspecific hybridization and introgression.
3. Results, first section – I think it would make sense to first write about the measurements of membrane lipid composition, and only then write about the phenotypic assays.
4. Figures: the font size is extremely small, especially the superscripts. I understand that space is limited, but there is some space to increase the font size a bit.
5. Figure S1 – I miss the lipid profile of *S. pombe* – this would in my opinion better show how deleting SHC1 changes the lipid profile (and brings it closer to *S. pombe*?)
6. Page 8 -Any idea why *japonicus* succeeds in growing without any unsaturated fatty acids / addition of Tween, while *S. pombe* cannot? (Maybe elaborate in discussion...)
7. Paragraph “asymmetrical saturated lipids...” – a bit more background to explain the rationale and what can be expected would be helpful (i.e. what is exactly meant with phase separation, what is disorder and how is it expected to be in wild-type cells... ?)
8. Do you have any hypothesis about how *japonicus* succeeded in adapting its level of asymmetric glycerophospholipids? Is this easily tuned by for example expression levels of certain genes, or rather due to more profound changes in enzymes and pathways?

Kevin Verstrepen

Reviewer #3

(Remarks to the Author)

In this manuscript, authors addressed how ergosterol and hopanoid pathways both play a role in lipid homeostasis in *S. japonicus*.

Overall, it is a useful study, but it lacks some aspects, hence I have a few suggestions.

First, the title does not really reflect the work in the manuscript. Although the title is about horizontal gene transfer, there is no experiment in this direction (although the general concept might be supported by the data). Therefore, I recommend the authors to reconsider their title. It is important that the audience of interest can read the title and get interested in the paper. Currently, it sounds too abstract.

The main experimental finding of the paper is that Erg and Hop pathways are working together and one can affect the other. And both of them can contribute to the lipid remodeling which can affect domains etc. In this sense, it is a bit convoluted to see the real function of these lipids. For example, Cryo-EM (Fig 3) results are interesting, but I do not get the reason for it. Authors state that they want to rule out the probe effect, but there is not a notable effect if you use them in low concentrations as known in the field. It seems like a very elaborate way of showing sterols and hopanoids contribute to the phase separation (which was shown previously as the authors also cited). To me, it would be a lot more interesting to have the cryo-EM data from vesicles directly-derived from the *S. japonicus* to test whether they have domains as a function of hopanoids or sterols. This would show the function of these lipids in the organism rather than in liposomes and vesicles which were both shown in the last decade by many researchers.

The claim “*S. japonicus* relies on hopanoid synthesis to survive in the absence of ergosterol” is shown with Erg1 inhibition with drugs in *shc1Δ* mutant. Did the authors confirm this genetically, eg, with double mutants whether they survive? This is not essential but would be nice to have to rule out any off-target effect of the drug used.

Minor:

Bar graphs should be more informative, eg, replicated and/or data points should be shown. Also, the colors should be made color-blind friendly.

Fig 1e -YES and YNB can be spelled out instead of abbreviation to make the figure more readable directly without reading any main text.

Version 1:

Reviewer comments:

Reviewer #1

(Remarks to the Author)

The authors have addressed all of my concerns. I think the manuscript was very nicely revised, and improved in many ways. It's an excellent study and I look forward to seeing it in print.

Reviewer #2

(Remarks to the Author)

I already liked the original version of the paper; the revised manuscript addresses all the concerns and questions that I raised. I also appreciate the additional work that was carried out in response to some of the other reviewer's suggestions, in particular the live cell analyses.

I have no further suggestions.

Kevin

Reviewer #3

(Remarks to the Author)

Authors addressed my concerns, and I recommend the manuscript for publication.

We are very grateful to the reviewers for their comments and suggestions. We believe that the review process has resulted in a much-improved manuscript.

Please note that in addition to incorporating the new data, and other edits addressing the reviewers' comments, we have changed the following:

- 1) In response to a request by the Reviewer 3 we changed the title to a more specific version. The new title is "**Horizontal acquisition of prokaryotic hopanoid biosynthesis reorganizes membrane physiology driving lifestyle innovation in a eukaryote**".
- 2) We shortened the abstract to under 150 words, and introduced other formatting edits, as required by *Nature Communications* guidelines.
- 3) We have changed the order of experiments shown in Fig. 1 and Supplementary Fig. 1, in response to a comment from the Reviewer 2, so that we now first show the lipidomics results and then describe the cell biological changes associated with the lack of either ergosterol or hopanoids in *S. japonicus*.
- 4) To streamline the flow of the manuscript and make space for new data, we have moved old Fig. 4f and 4g to Supplementary Fig 4e, f, and an edited version of old Fig. 2f to Supplementary Fig. 2e.
- 5) We have realized that in our original submission we made a calculation error affecting absolute values of triterpenoids detected by GC-MS. Specifically, we did not take into account the whole sample amount versus the derivatized portion during OD normalization. We corrected the numerical values (lines 299-300, Fig. 1e, Fig. 4b and Fig. 4f, and Supplementary Data 1, Tables 1-3) in the revised version. The relative tendencies and our conclusions remain unchanged.
- 6) The text outlining new data and additional discussion points requested by the reviewers is highlighted in red in the revised version of the manuscript.

Please find below the individual comments followed by our responses.

REVIEWER COMMENTS

Reviewer #1 (Remarks to the Author):

This study investigates the impact of horizontal gene transfer on lipid metabolism and cell physiology. The authors explore the role of squalene-hopene cyclase (Shc1) in hopanoid biosynthesis in *Schizosaccharomyces japonicus* and assess its functional integration when introduced into *S. pombe*, a related species that lacks hopanoids. By comparing lipid compositions and growth dynamics in both species, the authors demonstrate how Shc1 enables *S. pombe* to withstand anoxia and high temperatures. They attribute these resilience traits to changes in membrane properties and further validate findings using synthetic membrane models. This work reveals how a single gene transfer can induce physiological changes in growth and membrane composition, with potential implications for cell membrane biology and evolution.

However, the manuscript requires further clarification of the physiological and biophysical relevance of several key findings. Specifically, the data on diplopterol's influence on membrane order and phase behavior lacks sufficient discussion of previous studies and the significance of phase separation. Additionally, certain experiments involving membrane property probes would benefit from improved controls and interpretation. Seeking input from a membrane biophysicist would enhance the framing of these results. Below are detailed comments on major and minor points.

Major Points

1. **Lipidome Comparison:** The manuscript characterizes membrane lipid compositions in *S. japonicus* and *S. pombe* when either hopanoids, sterols, or both are present. However, it lacks a clear comparative analysis of how these lipid trends converge or diverge between the two species, particularly in the context of the synthetic membrane system used.

Thank you very much for your helpful comments and prompting us to do further experiments! We have now incorporated additional data highlighting the convergence of lipidome trends. Specifically, we have analysed the lipidomes of *S. pombe* wild-type and *ptdh1:shc1^{S.j.}-sfGFP* strains cultured under both normoxic and anoxic conditions. The triterpenoid profile was obtained by GC-MS (**new Fig. 4i**) and the overall lipidome analysis was performed using ESI-MS (**new Supplementary Fig. 4i-k**). To enable *S. pombe* growth in anoxia, we supplemented the medium with an unsaturated fatty acid source, Tween 80.

Our findings reveal that under oxygen-deprived conditions, *S. pombe* increases the synthesis of asymmetrical glycerophospholipids (GPLs) (**new Supplementary Fig. 4i**). This trend aligns with the lipidomic profile of *S. japonicus*, where the production of asymmetrical GPLs is inherently high under normoxia and increases further when cells are grown in anoxia (**now shown in Supplementary Fig. 1g**). These results suggest that in conditions where fatty acid desaturation is not possible due to the absence of oxygen, cells may preferentially synthesize asymmetrical GPLs to maintain membrane integrity and function. Our findings are in line with older reports showing that the budding yeast *S. cerevisiae* produces C12:0- and C10:0-containing asymmetrical GPLs in anoxia (PMID: 14063287). Importantly, our GC-MS analysis demonstrates that in anoxic conditions — where Erg1 activity is physiologically limited and ergosterol biosynthesis is disabled — Shc1^{S.j.}-expressing *S. pombe* cells increase hopanoid production (**new Fig. 4i**). In line with this, hopanoid synthesis is also increased in *S. japonicus* cells grown in the absence of oxygen (**Fig. 1e**). Thus, hopanoid biosynthesis is favoured when sterols cannot be synthesized.

These insights have been incorporated into the revised version of the manuscript (lines 359-378 of Results and lines 422-425 of Discussion).

2. **Live Cell Membrane Measurements:** This study would provide much deeper insight if the predicted impact of hopanoids or sterols on membrane properties were substantiated by direct measurements in live cells or membrane extracts, rather than inferred solely from model membrane systems.

This is a great suggestion, thank you! To address this question, we have carried out confocal microscopy-based measurements of membrane order in live fission yeast cells (*S. pombe* wild-type, and *S. japonicus* wild-type, *shc1Δ* and *erg1Δ* strains) labelled with C-laurdan. Our new results (**new Fig. 2h, i**) show that plasma membrane and nuclear membrane order in *S. japonicus* cells is higher relative to *S. pombe*, in line with our previous work (PMID: 31956022). Importantly, the genetic disruption of either sterol or hopanoid biosynthesis leads to a decrease in membrane order, indicating that both triterpenoids contribute to maintaining membrane order in *S. japonicus*. The new results are outlined in the revised version of the manuscript (lines 235-242).

3. **Gene Regulation Analysis:** Introducing Shc1 likely alters gene regulation of the whole cell, which could contribute to observed metabolite changes. A more comprehensive analysis or prediction of pathway regulation is necessary to elucidate these regulatory shifts.

We thank the reviewer for this suggestion. We have now probed potential changes in mRNA abundance in *S. pombe ptdh1:shc1^{S.j.}-sfGFP* cells versus *wild type* cells by RNAseq. Our analysis identified 214 differentially expressed genes, including 121 upregulated and 93

downregulated genes out of 5,093 detected transcripts (essentially covering the entire genome). To answer the reviewer’s question, **our transcriptomic data did not reveal significant differences in the expression of lipid metabolic genes, which could account for lipid metabolite changes** observed in *S. pombe* Shc1-expressing cells.

Interestingly, among the upregulated genes, we observed significant enrichment for the Gene Ontology (GO) term “oxidoreductase activity” (Fig. R1a). Among this group, there are several genes linked to cellular detoxification pathways. This observation suggests that *S. pombe* cells expressing Shc1^{S*j*} may experience oxidative stress, likely due to changes in membrane lipid composition. Among the significantly downregulated genes, we detected enrichment of the GO term “transmembrane transporter activity” (Fig. R1a), suggesting that the presence of hopanoids may elicit a membrane-associated regulatory response. Indeed, we found that *S. pombe* cells expressing Shc1^{S*j*} show sensitivity to tunicamycin, an inducer of ER stress, although to a lower extent than the mutant lacking *ire1*, an essential component of the RIDD-type unfolded protein response in fission yeast (Fig. R1b).

Figure R1. (a) Differentially expressed genes are grouped under two functional categories in *S. pombe* cells expressing Shc1^{S*j*} as compared to the wild type. Cells were grown in the modified minimal YNB medium at 30°C. AnGeLi suite was used for GO annotation. (b) Serial dilution assay of *S. pombe* wild type (WT), Shc1^{S*j*}-expressing and *ire1Δ* strains carried out in YES medium in the absence or presence of Tunicamycin.

Given that we did not observe changes in transcript abundance of lipid metabolic genes, which could readily explain lipid metabolite changes upon Shc1 expression, and the amount of data already in the paper, we prefer to leave this dataset out of the revised version of the manuscript. We will be exploring the physiological meaning of these results in our future work. Again, thank you for prompting us to do this experiment!

- 4. Phase Separation and Physiological Relevance:** The section exploring diplopterol’s role in promoting phase separation in giant unilamellar vesicles (GUVs) composed of yeast-like lipids is promising but lacks critical context. The manuscript does not reference foundational studies, such as Saenz et al. (PNAS, 2012: <https://www.pnas.org/doi/10.1073/pnas.1212141109>), which first demonstrated diplopterol’s ability to induce lo/ld phase separation and form an ordered phase with saturated lipids. The authors should discuss the biological significance of membrane phase separation, drawing on insights from studies involving GUVs, GPMVs, and observations in yeast and mammalian cells.

Apologies for not citing this wonderful paper (PMID: 22893685) in the original version of our manuscript – we did mention these findings in Discussion (“For instance, diplopterol has been

shown to support phase separation in the mixtures with a synthetic sphingolipid and DOPC”) but have inadvertently introduced a wrong citation during formatting! We thank the reviewer for their suggestions, which helped in improving the flow of the manuscript. We have added a statement explaining the biological significance of membrane phase separation observed in vitro to the Results chapter and cited a nice review (lines 214-217).

5. **FASTDiI Staining and Membrane Phase Interpretation:** “When combined with 30 mol% of either ergosterol or diplopterol, all three glycerophospholipids formed disordered membranes, as indicated by the labeling of GUVs with FASTDiI.” Staining with fastDiI is not a diagnostic indicator of whether a membrane is lo or ld, unless you are staining a phase separated vesicle where both lo and ld phases are present. The dye will preferentially partition in the presence of two phases, but it can still partition into mono-phase vesicles that are disordered. If referring to single-phase GUVs, alternative measurements such as DPH anisotropy or laurdan GP should be used to support these conclusions.

Thank you very much for pointing this out. We have modified the text, referring to single-phase GUVs (line 186). We moved DPH and C-laurdan measurements of two-component single-phase model membranes to Supplementary Fig. 2e, and Fig. 2b and Supplementary Fig. 2f, g, respectively.

6. **Laurdan GP and DPH Anisotropy in Phase-Separated Systems:** Laurdan GP and DPH anisotropy values in three-component GUVs (Fig. 2d-f) do not provide reliable insights here, as fluorescence probes can partition differently between coexisting phases. These values will not report an ‘average’ of membrane properties if they are phase separated, and should be removed. To validate claims, the authors should measure laurdan GP or DPH anisotropy with microscopy to discern phase-specific differences.

Again, thank you! We have now performed phase-specific membrane order measurements using C-laurdan. Our data indicate a co-existence of two phases in the following three component mixtures: POPC-DPPC-Ergosterol, SDPC-DPPC-Ergosterol and SDPC-DPPC-Diplopterol (**new Fig. 2e-g and Supplementary Fig. 2h**). In line with data shown in **Fig. 2d**, we have observed a higher ordering potential for ergosterol as compared to diplopterol. We edited the text accordingly (lines 229-234 in the revised version of the manuscript).

Of note, we have removed the data on spectroscopic DPH anisotropy measurements in three-component membranes from the revised version of the manuscript, as prompted by the Reviewer.

Minor Points

1. **Triterpenoid Pathway Disruption and Membrane Integrity:** The manuscript should discuss the potential impact of squalene accumulation on membrane integrity, especially considering the reduced growth rate of the constitutive *prg3* *S. pombe* strain at high temperatures. I recall there is at least one published study that has demonstrated squalene accumulation can destabilize membranes in yeast (for example: <https://pubmed.ncbi.nlm.nih.gov/22342273/>).

Again, many thanks for this suggestion! We now mention a possibility that squalene accumulation may impact membrane integrity and cite this paper alongside another report (lines 350-351). Of note, Shc1 expression in *S. pombe* also rescues the growth of the wild-type strain at 37°C (**Fig. 4g and Supplementary Fig. 4h**).

2. **Use of Constitutive Strain in *S. pombe*:** The authors chose a constitutive *S. pombe* strain rather than using terbinafine to modulate ergosterol levels, as in *S. japonicus*. An explanation for this approach would enhance the reader’s understanding.

We decided to use a constitutive mutant to have a better control over experimental conditions. Importantly, drugs come with risks of off-target effects – e.g., it has been suggested that the translation initiation factor Tif302 in *S. pombe* could be an off-target for terbinafine (PMID: 33223513). We added a sentence outlining our rationale to the text (lines 328-329).

3. **Anaerobic Growth of *S. japonicus*:** The morphological differences observed in *S. japonicus* under anaerobic conditions without ergosterol supplementation should be discussed, as this could provide insights into hopanoid functionality. Please discuss how *S. japonicus* morphology in anaerobic conditions differ without access to ergosterol.

To address your question, we have now included measurements of cell length and width at division of *S. japonicus wild type* strain grown in anoxia (**new Fig. 1h-j**). We observe a mild deregulation of cell morphology, as compared with “normal”, aerobic growth. Please see lines 162-165 of the revised version of the manuscript.

4. **Explanation of SoamD Score:** The results section should briefly explain the SoamD score and its application, as this metric is not widely known among readers.

The SoamD score was originally explained in Supplementary Fig. 1 legend, but we have now included the description in the Results – please see lines 124-125 of the revised version of the manuscript.

5. **Unsupported Claims:** The sentence beginning with “For instance, diplopterol ...” describes phase separation with diplopterol, sphingolipid, and DOPC, however, this was not supported by the corresponding citation.

Apologies for this! We have inadvertently lost the correct citation (PMID: 22893685) during formatting. Corrected!

6. **Integration of SHC and Fatty Acid Desaturation:** The claim that fatty acid desaturation facilitates SHC integration into membranes lacks citation. This claim was not supported by the corresponding citation. Supporting evidence for this assertion is needed to substantiate it. Please double check the accuracy of all citations.

We did not intend to claim that fatty acid desaturation facilitates SHC integration into membranes. Rather, we speculated that increasing fatty acid desaturation might help to accommodate hopanoids in membranes, and in support of this hypothesis, we cited a report demonstrating that *Tetrahymena pyriformis* cells exhibit higher GPL desaturation when tetrahymanol and diplopterol are the main membrane triterpenoids, as compared to ergosterol (PMID: 109555). We have now edited this part to clarify the message (lines 435-436 in the revised manuscript). Based on this finding in *Tetrahymena pyriformis*, another organism with abundant hopanoids, and our own results showing that the heterologous expression of Shc1 in *S. pombe* leads to an increase in FA desaturation (**Fig. 4d**), we argue that FA desaturation and FA asymmetry both might facilitate the accommodation of hopanoids in eukaryotic membranes.

7. Please include line numbering in future revisions to help reviewers describe where in the manuscript they have concerns.

Yes, of course – included!

Reviewer #2 (Remarks to the Author):

This beautiful study explores how horizontal transfer of a bacterial squalene-hopene synthase gene, SHC1, into the genome of *S. japonicus* allowed this yeast to adapt to growth in

complete anaerobiosis by avoiding the need to produce the normal sterols that require oxygen for their synthesis. The authors show how hopanoids can replace sterols both in vivo as well as in vitro (artificial model membranes) on the condition that there is also an increase in asymmetrical glycerophospholipids in the membranes. Lastly, they mimic the horizontal gene transfer event by introducing SHC1 into the genome of *S. pombe*, a yeast related to *S. japonicus* but lacking its capacity to grow in anoxia. This last set of experiments confirms that SHC1 expression in *S. pombe* indeed confers better growth in anoxia, although supplementation with unsaturated fatty acids was still necessary. In addition, expression of SHC1 also bestowed *S. pombe* with a tolerance to higher temperatures, mimicking the temperature tolerance of *S. japonicus*.

Together, this study unveils the molecular details of how horizontal gene transfer bestows important new capacities onto the receiving organism, but at the meantime also requires further adaptation and tuning of the receiving organism's general physiology. It is exactly this last part that makes the study innovative and interesting. Plus, apart from helping to understand the evolutionary processes related to horizontal gene transfer, the study also adds to our understanding of anaerobic growth and lipids in general.

The paper is beautifully written and, despite its multi-disciplinary nature easy to follow even for non-experts. That said, since I am not an expert in lipid metabolism, I cannot comment in detail on the scientific rigor of the results regarding lipid composition in general, and the artificial membranes in particular. Instead, I focus on the molecular-evolutionary part. I only have some minor questions and suggestions.

1. In the introduction, it could be interesting to mention whether there are other known cases of organisms adapting to anoxia, and how they managed (are there other cases of SHC, as well as other cases of HGT that required further tuning of the receiving host organism).

Thank you for your supportive review! We now mention other instances of horizontal transfer of genes encoding the squalene hopene cyclases in the Introduction (lines 61-62). We added further discussion on other HGT events linked to the adaptation to anaerobic lifestyle to the Discussion (lines 468-474).

2. It could be interesting to mention in the intro and/or discussion that the general principles found here may not only apply to horizontal gene transfer, but also to interspecific hybridization and introgression.

It's a great idea. We now mention that similar principles may apply to interspecific hybridization and introgression in the Discussion (lines 483-485).

3. Results, first section – I think it would make sense to first write about the measurements of membrane lipid composition, and only then write about the phenotypic assays.

We have now swapped the order, and we feel that it does read better.

4. Figures: the font size is extremely small, especially the superscripts. I understand that space is limited, but there is some space to increase the font size a bit.

We have increased the font size for all figures.

5. Figure S1 – I miss the lipid profile of *S. pombe* – this would in my opinion better show how deleting SHC1 changes the lipid profile (and brings it closer to *S. pombe*?)

The *S. pombe* and *S. japonicus* lipidomes were originally compared in our 2020 paper (PMID: 31956022). Here, we also show the *wild type S. pombe* lipid profile in **Fig. 4b-d** and **Supplementary Fig. 4a-e**, as a control for our experiments on introduction of Shc1 and

attenuating Erg1 expression. Deleting *shc1* does not bring the overall lipidome of *S. japonicus* closer to *S. pombe*, e.g., *S. japonicus* continues synthesizing large amounts of asymmetrical C10:0-containing glycerophospholipids (**Supplementary Fig. 1g**). That said, we see a fascinating convergence of lipidomic trends between *S. japonicus* and anaerobically grown Shc1-expressing *S. pombe* (new experiments performed in response to Point 1 of the Reviewer 1 (described in **new Fig. 4i** and **Supplementary Fig. 4i-k**). We copy our response to the Reviewer 1 below:

Our findings reveal that under oxygen-deprived conditions, *S. pombe* increases the synthesis of asymmetrical glycerophospholipids (GPLs) (**new Supplementary Fig. 4i**). This trend aligns with the lipidomic profile of *S. japonicus*, where the production of asymmetrical GPLs is inherently high under normoxia and increases further when cells are grown in anoxia (**now shown in Supplementary Fig. 1g**). These results suggest that in conditions where fatty acid desaturation is not possible due to the absence of oxygen, cells may preferentially synthesize asymmetrical GPLs to maintain membrane integrity and function. Our findings are in line with older reports showing that the budding yeast *S. cerevisiae* produces C12:0- and C10:0-containing asymmetrical GPLs in anoxia (PMID: 14063287). Importantly, our GC-MS analysis demonstrates that in anoxic conditions — where Erg1 activity is physiologically limited and ergosterol biosynthesis is disabled — Shc1^{S_j}-expressing *S. pombe* cells increase hopanoid production (**new Fig. 4i**). In line with this, hopanoid synthesis is also increased in *S. japonicus* cells grown in the absence of oxygen (**Fig. 1e**). Thus, hopanoid biosynthesis is favoured when sterols cannot be synthesized.

These insights have been incorporated into the revised version of the manuscript (lines 359-378 of Results and lines 422-425 of Discussion).

6. Page 8 -Any idea why *japonicus* succeeds in growing without any unsaturated fatty acids / addition of Tween, while *S. pombe* cannot? (Maybe elaborate in discussion...)

Our data suggest that either increasing the proportion of asymmetrical glycerophospholipids or increasing FA desaturation may confer similar physicochemical properties to cellular membranes. For instance, **Fig. 2** and **Fig. 3** show that either symmetrical desaturated glycerophospholipids (POPC or DOPC), or the asymmetrical saturated SDPC support membrane phase separation into liquid-ordered and liquid-disordered domains in three component mixtures containing the gel-like saturated lipid DPPC. Glycerophospholipid asymmetry is prevalent in the lipidome of *S. japonicus* cells grown in normoxia, and the synthesis of these lipids increases further in anoxia. Of note, *S. pombe* and *S. cerevisiae* are capable of synthesizing asymmetrical lipids even in normoxia (PMID: 31956022 and PMID: 32850859), and increase their synthesis further under oxygen limitation (**new Supplementary Fig. 4i** of this manuscript and PMID: 14063287). Thus, it appears that the synthesis of medium chain fatty acids leading to the production of asymmetrical glycerophospholipids could be a regulatable property in other organisms, but it became fixed in the *S. japonicus* lineage. That said, our experiments show that *S. pombe* cannot solely rely on asymmetrical glycerophospholipids to maintain its membrane function in anoxia, hence the need for supplementation with unsaturated fatty acids (**Fig. 4h, i** and **Supplementary Fig. 4i-k**).

7. Paragraph “asymmetrical saturated lipids...” – a bit more background to explain the rationale and what can be expected would be helpful (i.e. what is exactly meant with phase separation, what is disorder and how is it expected to be in wild-type cells... ?)

Thank you for pointing this out! We have edited the manuscript – hopefully the rationale is well explained. We included the following sentence to explain how lateral phase separation in model membranes can relate to membrane biology: “Lipid-lipid interactions resulting in macroscopic phase separation in model membranes and plasma membrane-derived vesicles

are thought to contribute to the generation and/or stabilization of lateral heterogeneities in vivo, critical for the function of biological membranes” (lines 214-217 of the revised manuscript). We have also provided a nice review covering the subject (PMID: 29410529).

8. Do you have any hypothesis about how *japonicus* succeeded in adapting its level of asymmetric glycerophospholipids? Is this easily tuned by for example expression levels of certain genes, or rather due to more profound changes in enzymes and pathways?

This is a very interesting question! We have discussed several possibilities in our 2020 *Current Biology* paper. Copying from PMID: 31956022:

“*S. japonicus* FAS exhibits overall sequence conservation with other fungal FA synthases (see STAR Methods for corrected *fas1* ORF), although it does have a number of potentially interesting substitutions to be analyzed in future. The budding yeast mutagenesis experiments suggest that the evolutionary acquisition of new FAS functionality may not necessarily require a large number of steps [56]. Importantly, other cellular factors may influence the spectrum of FAS products. An interesting possibility could be the different ratios between acetyl- and malonyl-CoA in *S. japonicus* and *S. pombe*. Relative depletion of malonyl-CoA results in the enzyme favoring priming over elongation [56] and increases the production of shorter FAs, whereas hyperactivation of the enzyme responsible for malonyl-CoA synthesis shifts FA distribution toward longer lengths [57]. Curiously, the mRNA for this enzyme, *cut6* [58], is significantly upregulated in *S. pombe* cells expressing *S. japonicus* FAS (Tables vii and viii in Data S1), perhaps as a part of a Mga2-based compensation mechanism triggered by elevated C10:0 levels. Other cellular adaptations are likely required to efficiently handle C10:0 alongside LCFAs, including modifications of lipid remodeling enzymes.”

We have also included a short discussion on this topic in the Discussion of this manuscript (lines 425-432).

Reviewer #3 (Remarks to the Author):

In this manuscript, authors addressed how ergosterol and hopanoid pathways both play a role in lipid homeostasis in *S. japonicus*. Overall, it is a useful study, but it lacks some aspects, hence I have a few suggestions.

First, the title does not really reflect the work in the manuscript. Although the title is about horizontal gene transfer, there is no experiment in this direction (although the general concept might be supported by the data). Therefore, I recommend the authors to reconsider their title. It is important that the audience of interest can read the title and get interested in the paper. Currently, it sounds too abstract.

Thank you for all your suggestions!

To address your first point, independent instances of horizontal transfer of SHC and SHC-related genes have been well documented in the literature (e.g., PMID: 34353908, PMID: 19207562). We agree that we should provide a more specific title. As mentioned above, we have now changed the title to: **Horizontal acquisition of prokaryotic hopanoid biosynthesis reorganizes membrane physiology driving lifestyle innovation in a eukaryote.**

The main experimental finding of the paper is that Erg and Hop pathways are working together and one can affect the other. And both of them can contribute to the lipid remodeling which can affect domains etc. In this sense, it is a bit convoluted to see the real function of these lipids.

For example, Cryo-EM (Fig 3) results are interesting, but I do not get the reason for it. Authors state that they want to rule out the probe effect, but there is not a notable effect if you use them in low concentrations as known in the field. It seems like a very elaborate way of showing sterols and hopanoids contribute to the phase separation (which was shown previously as the authors also cited). To me, it would be a lot more interesting to have the cryo-EM data from vesicles directly-derived from the *S. japonicus* to test whether they have domains as a function of hopanoids or sterols. This would show the function of these lipids in the organism rather than in liposomes and vesicles which were both shown in the last decade by many researchers.

We were probably too brief introducing our cryo-EM measurements in Results in the original version of the manuscript, although we discussed their importance in Discussion! They do not simply verify our findings with fluorescent probes. First, cryo-EM has allowed us to measure the thickness of membranes made with asymmetrical saturated glycerophospholipids, as compared to more “standard” symmetrical unsaturated glycerophospholipids. Our finding that asymmetrical lipids form thinner membranes is important, because it provides a biophysical explanation for the shortening of transmembrane helices in a subset of *S. japonicus* transmembrane proteins as compared to their orthologs from other fission yeasts (PMID: 31956022). It might have broader implications for understanding the organelle-dependent sorting of transmembrane domains (PMID: 20603021). Second, cryo-EM has allowed us to visualize the phase-separated domains at high resolution and quantify domain thickness in three-component phase-separated membranes containing either ergosterol or diplopterol. We show that domains formed in mixtures of SDPC and DPPC with either ergosterol or diplopterol exhibit comparable values of hydrophobic thickness, suggesting that either triterpenoid is able to accommodate similar sets of proteins and support membrane functions in vivo (**Fig. 3d-f**). We have edited the manuscript to give a proper rationale for our cryo-EM experiments (lines 253-254) and highlight the biological significance of our findings (lines 278-280).

To address your second point, whether it would be useful to perform cryo-EM on vesicles directly derived from *S. japonicus* cells, we believe that interpretation of the results would be very difficult. The presence of the thick wall in yeast cells requires harsh and prolonged procedures to purify organelles, potentially affecting membrane architecture. Making GUVs from the mixtures of TLC-purified cellular polar lipids is also not ideal since the composition of these mixtures is difficult to control. Essentially, for any observation we would not be able to correlate an image measurement with a specific lipid composition as we do for model membranes. We plan to eventually perform cryo-EM/cryo-ET measurements directly in cell sections prepared by FIB-milling, but this would be a large and independent project that we believe is beyond the scope of this study.

Thus, to address the roles of hopanoids and ergosterol in sustaining membrane properties in vivo, we have now measured membrane order in the wild type, *shc1Δ* and *erg1Δ* *S. japonicus* cells using C-laurdan (**new Fig. 2h, i**). We show that the lack of either hopanoids or ergosterol leads to a pronounced drop in the order of the nuclear membrane and the plasma membrane, indicating that both triterpenoids contribute to membrane organization in this fission yeast in vivo.

The claim “*S. japonicus* relies on hopanoid synthesis to survive in the absence of ergosterol” is shown with Erg1 inhibition with drugs in *shc1Δ* mutant. Did the authors confirm this genetically, eg, with double mutants whether they survive? This is not essential but would be nice to have to rule out any off-target effect of the drug used.

Thank you for this suggestion. We now include the genetic interaction data showing that the double *shc1Δ erg1Δ* mutant is not viable, in the new version of the manuscript (lines 102-106 of Results and lines 573-582 of Materials and Methods).

Additionally, the claim that *S. japonicus* relies on hopanoid synthesis in the absence of ergosterol is supported by the serial dilution assay that demonstrates that the *shc1Δ* mutant fails to grow in anoxic conditions (**Fig. 1c**) — a physiological scenario where ergosterol biosynthesis is inhibited, as confirmed by our GC-MS results (**Fig. 1e**).

Minor:

Bar graphs should be more informative, eg, replicated and/or data points should be shown. Also, the colors should be made color-blind friendly.

Thank you! We have now modified the graphs according to your suggestions.

Fig 1e -YES and YNB can be spelled out instead of abbreviation to make the figure more readable directly without reading any main text.

Thanks! New Fig. 1h now includes the spelled-out forms of YES and YNB.

Reviewer 1

The authors have addressed all of my concerns. I think the manuscript was very nicely revised, and improved in many ways. It's an excellent study and I look forward to seeing it in print.

Thank you again for your helpful suggestions for revising the paper!

Reviewer 2

I already liked the original version of the paper; the revised manuscript addresses all the concerns and questions that I raised. I also appreciate the additional work that was carried out in response to some of the other reviewer's suggestions, in particular the live cell analyses. I have not further suggestions. kevin

Thank you very much for your kind comments!

Reviewer 3

Authors addressed my concerns, and I recommend the manuscript for publication.

Thank you!